# Influence of Maxwell Stiffness in Damage Control and Analysis of Structures with Added Viscous Dampers

**Jorge Conde *** and **Alejandro Bernabeu**

Escuela Técnica Superior de Arquitectura, Universidad Politécnica de Madrid, Av. Juan de Herrera 4., 28040 Madrid, Spain; alejandro.bernabeu@upm.es
* Correspondence: jorge.conde@upm.es; Tel.: +34-680128040

**Featured Application: In the absence of more accurate data, stiffness of added viscous damping systems based on brace extender properties should be substantially reduced for analysis. Factors between 0.25 and 0.50 are recommended.**

**Abstract:** Viscous damping systems are often implemented in structures to reduce seismic damage. The stiffness of these elements is dominated by the most flexible part of the set including brace extender, auxiliary mounting elements and damping unit. Existing experimental data are used in this study to show that the actual stiffness of the set is about 25% to 50% of the value generally adopted in current engineering practice, which is based solely on the brace extender. A numerical study shows that this reduction has large implications for several variables related to damage control: residual drift ratio, storey acceleration and plastic strain energy dissipated by the frame members. Other variables, such as member forces and rotations, can experience large variations, particularly for non-linear dampers and high damping levels, especially in the top part of the building and more conspicuously for moderate earthquake intensities. In the absence of accurate data, Maxwell stiffness for analysis based on brace extender properties should be substantially reduced, with recommended factors between 0.25 and 0.50. Given the scarcity of experimental data, these results should be considered preliminary.

**Keywords:** viscous dampers; Maxwell stiffness; damage control

## 1. Introduction

During the last years of the past century, several seismic events (i.e., 1994, Northridge, USA; 1995, Kobe, Japan) took place that substantially shifted the focus of earthquake engineering [1,2]. Even though many of the buildings involved were designed according to modern seismic codes, non-structural damage resulted in enormous economic losses and traumatic post-event recovery, making the engineering community realize that enhanced seismic performance, beyond the mere life-safety level, was necessary [3,4]. Recent seismic events (2010, Chile; 2011, Christchurch, New Zealand) have thrust the need for even more demanding objectives, with emphasis on structural reparability, damage-protection and resilience [5–7]. In this regard, declassification of the military viscous damper technology (developed during the cold war) during the 1990s allowed application of these devices to vibration and earthquake protection of civil structures [8]. Addition of viscous dampers to structures is beneficial in several ways: first, seismic energy input is dissipated through viscous mechanisms rather than structural hysteresis, thus providing an effective protection to the main structural elements [9,10]; second, interstorey drift, peak storey acceleration and peak velocity are reduced, preventing damage to non-structural elements [11,12]. For these reasons, viscous dampers have been widely used to provide added seismic resilience to structures [13–15]. The Japanese case is particularly illustrative: The Japan Society of Seismic Isolation (JSSI) was founded in 1993, and, soon after, the 1995 Kobe earthquake

boosted the application of passive protective technologies in this country [16]; by 2011, supplemental damping systems had been already validated with full-scale tests [17] and around 1000 buildings in Japan already counted with passive protection systems [18]; during the 2011 Tohoku-Oki Earthquake, 11 buildings equipped with these types of systems were instrumented with sensors; the recorded signals and post-earthquake inspection showed their excellent seismic behavior [19]. The first codified provisions for supplemental damping systems were developed in the US [20]. Revision of the European seismic code currently under work contemplates inclusion of clauses relative to these systems [21].

The behavior of these devices has been thoroughly studied [22–29]. A fractional derivative model has been proposed as the most accurate mathematical representation. For practical purposes, this model has often been simplified to a pure dashpot. Added viscous damping systems can be modeled by the serial combination of a spring and a dashpot, where the former represents the flexibility of the different elements linking the pure viscous element to the main structure, and the damper flexibility itself; this last term is often neglected. For a very rigid spring, the system can be assimilated to a pure dashpot, and a closed-form solution under harmonic load is possible; this conveniently allows for the use of simplified methods, such as response spectrum analysis (RSA) [30] implemented in ASCE/SEI 7–16 [31]. However, experimental [32,33] and analytical [34] studies have shown that inclusion of damping system flexibility is essential in achieving an accurate approximation to the dynamic structural response of systems equipped with added viscous damping. Moreover, a minimum level of stiffness is required for adequate performance of the damping system, as it has been shown that excessive flexibility leads to both substantial reduction of dissipated energy and appearance of damper forces in phase with structural restoring forces, the latter being detrimental for the main structure. Rules have been proposed to estimate the minimum amount of stiffness required to achieve a reasonable level of system efficiency [35,36]. However, they are usually based on the behavior of Single-Degree-Of-Freedom (SDOF) systems. One of the objectives of this study is to investigate their adequacy for Multi-Degree-Of-Freedom (MDOF) systems.

If the stiffness of the auxiliary components of the bracing system is included in the analysis model, it is common practice to estimate its value based only on the brace extender cross section and length, on the assumption that this approach will render a suitable approximation to the system properties [8]. The main reason for this simplification is the absence of accurate data regarding the stiffness of other auxiliary elements (such as gusset plates, cleats, bolts, clevises, frontal plates, etc.) included in the set. Moreover, the stiffness component of the damper element itself is generally unavailable in catalogs and manufacturer data, implying that this variable can be neglected by the designer. A second objective of this study is to examine the validity of this assumption, and to determine to which extent the stiffness of secondary elements—other than the brace extender and including the damper stiffness component—should be included in analysis models.

In order to achieve the study targets, this study comprises two parts. First, available experimental data are used to assess the range of influence of auxiliary elements on the overall flexibility of the added damping system. Second, a parametric study is carried out in which two frames equipped with different viscous damping systems are modeled with and without consideration of the brace elements and auxiliary system stiffness. The frame is subjected to sets of accelerograms under different intensities, and the peak values for relevant variables at different storeys are compared. The study shows that the errors incurred by considering only the brace extender stiffness (without inclusion of other auxiliary elements) are not negligible. In fact, for some variables and storeys, they are of the same order of magnitude than the errors incurred by totally neglecting the stiffness. Thus, an accurate result can only be obtained if the flexibility of all auxiliary elements (including the damper itself) are considered. Given the scarcity of experimental data, these results should be considered as preliminary. However, they suggest the need for future experimental characterization of complete damper units, including all mounting elements.

## 2. Materials and Methods

### 2.1. Mathematical Models

Figure 1 shows several mathematical models used to represent viscous damping systems. The pure dashpot (Figure 1a) is an extreme case of the Maxwell system in which the stiffness is infinite. The Maxwell system (Figure 1b) includes both linear stiffness and viscous damping and renders an accurate description of the damper and its mounting elements. The Kelvin model (Figure 1c) presents the spring and damper in parallel and has been proposed by several authors as a simple approximation to the more complex Maxwell system. The standard model (Figure 1d) includes a Maxwell system in parallel with a linear or non-linear spring and can be used to represent the viscous damper inserted within a structural system. In all cases, hysteretic loops for steady state response under harmonic excitation of amplitude $u_o$ and circular frequency $\omega$ are presented.

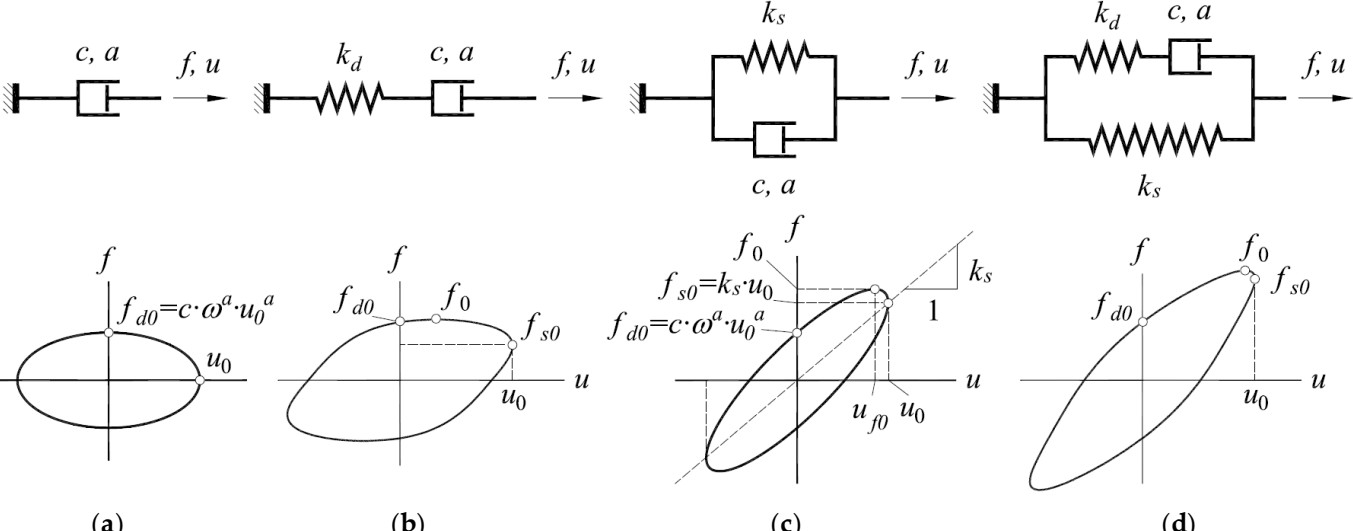

**Figure 1.** Mathematical models for viscous dampers and their steady-state hysteresis loops for forced harmonic excitation; (**a**) pure dashpot; (**b**) Maxwell model; (**c**) Kelvin model; (**d**) Standard solid model.

The constitutive equation of the pure dashpot is:

$$f = c|\dot{u}|^a sgn(\dot{u}) \tag{1}$$

where, $f$ indicates force, $u$ is displacement, $c$ is the damping coefficient, sgn($\cdot$) is the sign function and $a$ is the damper exponent, with $a = 1$ corresponding to linear dampers. For earthquake mitigation, exponents below 1 are common. The constitutive equation for the non-linear Maxwell model is:

$$\dot{u} = \frac{\dot{f}}{k_d} + \frac{|f|^{1/a}}{c^{1/a}} sgn(f), \tag{2}$$

where, $k_d$ is the damping system stiffness. If $k_d$ takes a very small value, the first term on the right side of Equation (2) prevails. Neglecting the second term and after integration, the equation reduces to that of a pure spring with small stiffness and negligible damping. If $k_d$ takes a very large value, the same term vanishes, and the equation reduces to a pure dashpot. These extreme cases show the behavior dependence of the Maxwell system on its stiffness. For linear dampers, Equation (2) can be expressed as

$$\dot{f} = k_d \dot{u} - \frac{f}{\lambda}; \tag{3}$$

where, $\lambda$ (=$c/k_d$), expressed in seconds, is referred to as relaxation time. For use in earthquake analysis, the behavior of the models is generally characterized for forced harmonic excitation on the assumption that, under earthquake excitation with broad frequency content, the structure vibrates mainly with its fundamental period. For forced harmonic excitation of circular frequency $\omega$ and amplitude $u_0$, the steady-state solution to Equation (3) is:

$$f = \beta_s k_d u_0 \sin(\omega t) + \beta_d c \dot{u}_0 \cos(\omega t), \tag{4}$$

$$\beta_s = \frac{\omega^2 \lambda^2}{1 + \omega^2 \lambda^2}, \tag{5}$$

$$\beta_d = \frac{1}{1 + \omega^2 \lambda^2}. \tag{6}$$

Equation (4) can be compared to the steady-state solution for the Kelvin system, which adopts the same form with $\beta_s = \beta_d = 1$. For this reason, the Kelvin system has been proposed as an approximation to the Maxwell system. $\beta_s$ and $\beta_d$ are dependent on excitation frequency and relaxation time and represent the relative magnitudes of spring and damper response, respectively. Figure 2 presents a plot of these coefficients for two different values of relaxation time as a function of the excitation period. For very short periods the predominant behavior is that of a pure spring, whereas for very long periods pure damping dominates the response. In between these two extremes, the behavior is a blend of damper and spring, resulting in the characteristic hysteretic shape of the Maxwell system. It is clear that for MDOF structures, the performance of the added damping system will be different for the fundamental mode and higher modes where spring-like, non-dissipative behavior is to be expected.

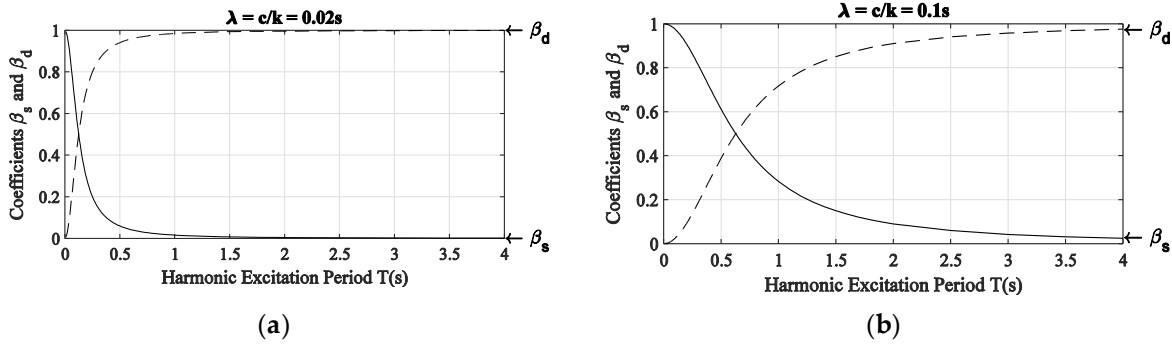

**Figure 2.** Coefficients $\beta_s$ and $\beta_d$ in Equation (4) as a function of harmonic excitation period $T$: (**a**) $\lambda$ = 0.02 s; (**b**) $\lambda$ = 0.1 s.

The energy dissipated in a cycle of harmonic motion for a pure damper is [26]:

$$W_d = \pi b c \omega^a u_0^{1+a}, \tag{7}$$

where, $b$ is a function of $a$ representing the relationship between dissipated energy in the non-linear damper and a linear damper with the same peak force response and amplitude. $b$ can be found as

$$b = \frac{2^{2+a}}{\pi} \frac{\Gamma^2(1 + a/2)}{\Gamma(2 + a)}; \tag{8}$$

in this expression $\Gamma(\cdot)$ represents the gamma function. For any non-linear damper, energy equivalence can be used to propose an equivalent linear damper with damping coefficient:

$$c_{1,eq} = b c \omega^{a-1} u_0^{a-1}, \tag{9}$$

and the relaxation time $\lambda_a$ for the non-linear Maxwell system can be defined as $c_{1,eq}/k_d$. For the standard solid model presented in Figure 1d, the viscous damping ratio $\zeta$ can then be expressed using Jacobsen's equation:

$$\zeta = \frac{W_d}{4\pi W_s};\qquad(10)$$

where, $W_s = k_s \cdot u_0^2/2$ is the strain energy at peak displacement. Combining the previous equations, and recalling that $\omega = 2\pi/T$:

$$\frac{k_s}{k_d} = \frac{\pi\lambda_a}{\zeta T}.\qquad(11)$$

Some authors recommend a maximum value of $k_s/k_d$ in order to keep the ratio $\lambda_a/T$ low, thus ensuring prevalence of the damper behavior. For instance, Lin and Chopra [35] suggested $k_s/k_d \leq 0.2$; with a 30% added damping ratio, this condition is transformed approximately into $\lambda/T \leq 0.02$, and the authors conclude that, if this condition is met, the influence of brace stiffness on response is negligible regardless of the damper exponent. Fu and Kasai [36] suggested a more restrictive limit $k_s/k_d \leq 0.1$; for the same viscous damping ratio, this limit results in $\lambda/T \leq 0.01$. The large difference between both limits indicates a certain ambiguity in this condition. Moreover, Equation (11) applies only to SDOF systems. Despite that, its use has been extended to MDOF systems. For this type of systems, however, it has been proposed that a Maxwell stiffness for all storeys $k_{d,n} = k_{s1}$ (first storey stiffness) is enough for a reasonable performance of the added damping system [37]. Verification of the validity of these conditions is of interest.

Figure 3 presents hysteresis loops for Maxwell models with different values of $k_d$, for the linear ($a = 1$) and non-linear ($a = 0.5$) cases, both subjected to the same forced harmonic excitation and energy-equivalent damping coefficients as per Equation (9). The figure shows the features of behavior with limited stiffness: (i) large reduction of dissipated energy expressed as area enclosed within the loop; (ii) appearance of forces at maximum displacement, i.e., increase in storage stiffness; (iii) reduction of forces at maximum velocity, i.e., decrease in loss stiffness; and (iv) moderate reduction of maximum forces.

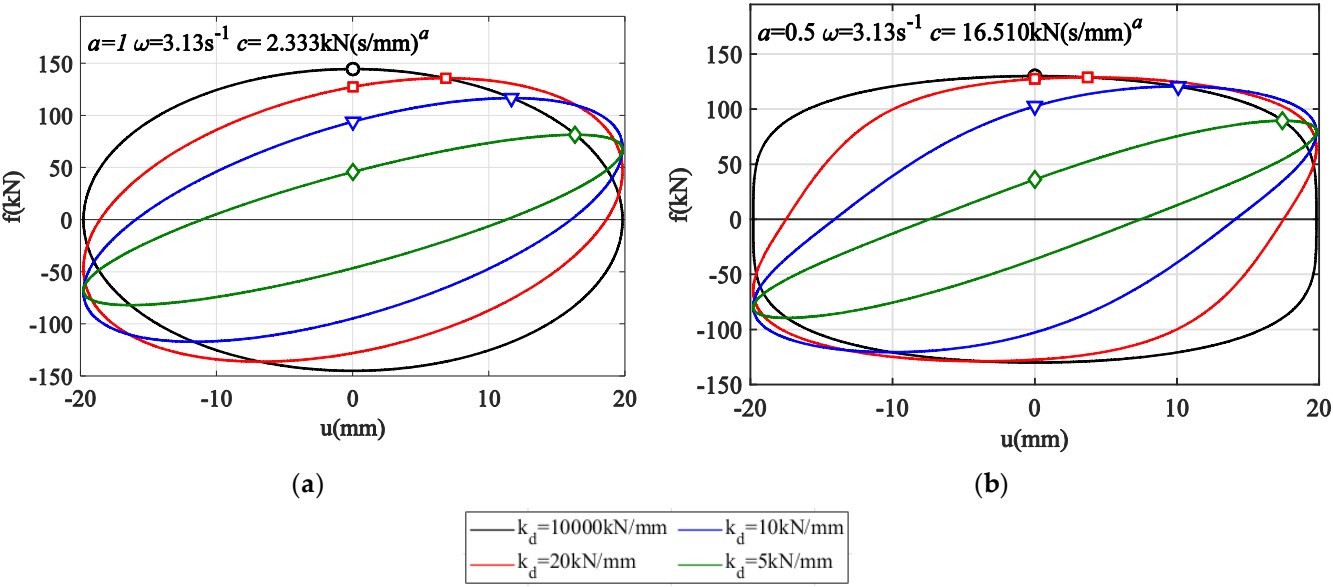

**Figure 3.** Hysteretic force-displacement loops for Maxwell systems subjected to forced harmonic vibration with different values of $k_d$ = 10,000 kN/mm, 20 kN/mm, 10 kN/mm, 5 kN/mm; (**a**) linear case; (**b**) non-linear case.

### 2.2. Estimate of Maxwell Stiffness

Viscous dampers are generally mounted using brace extenders and attached to the structure by means of strong gusset plates. Figure 4 shows a typical mounting arrangement for a single diagonal element. The stiffness and damping coefficient related to the damper axis are referred to as $k_b$ and $c_b$. A series of auxiliary elements (gusset plates, cleats, bolts, clevises, end plates, etc.) are needed in addition to the brace extender and damper. Elastic behavior is assumed for all elements in the set. The damper element itself presents a certain stiffness $k_{bd}$, which in typical application is assumed as infinite. The stiffness of other elements in the set is generally uncertain at the design stage, as many of these elements are typically provided by the damper manufacturer. For other elements, the stiffness can only be grossly approximated; for instance, the flexibility of joint elements can be estimated using the component-based approach in Eurocode 3, part 1–8 [38]. For this reason, the overall stiffness is often calculated using the brace extender cross section stiffness over the total centerline length. It is assumed that the combined stiffness of the set does not deviate significantly from this value. The model is suitable for seismic analysis not involving limit states of the device, for which a more sophisticated mathematical model has been developed [39].

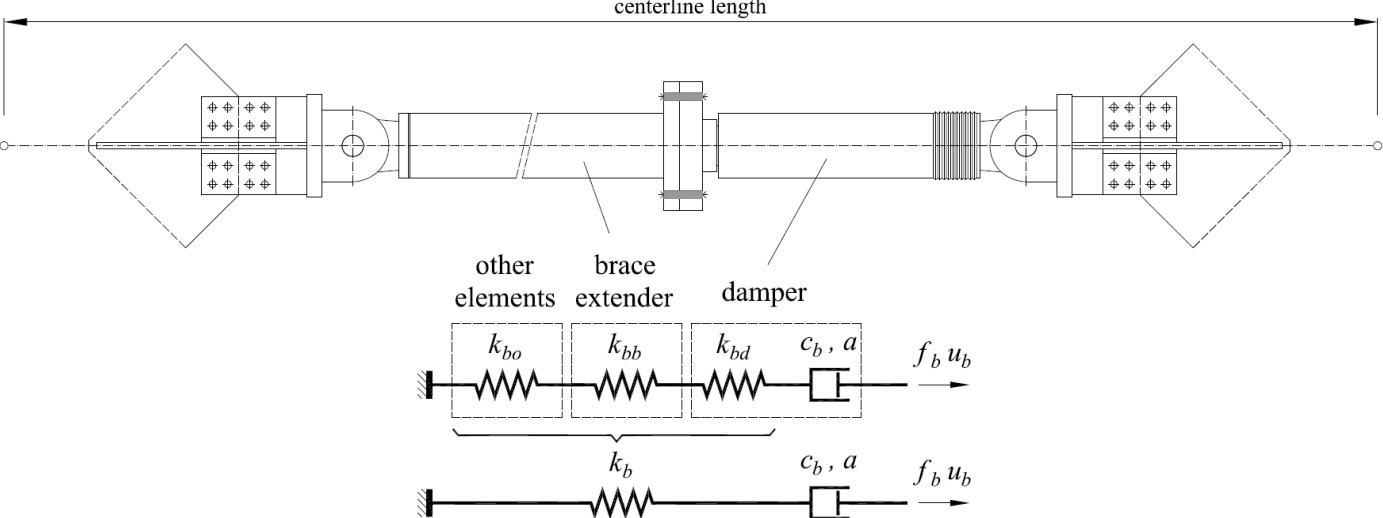

**Figure 4.** Stiffness of damper and auxiliary mounting elements.

The actual total damping system stiffness $k_b$ can be calculated as:

$$\frac{1}{k_b} = \frac{1}{k_{bo}} + \frac{1}{k_{bb}} + \frac{1}{k_{bd}}. \tag{12}$$

Figure 5 shows the resulting value of $k_b$ considering only two components ($k_{bb}$ and $k_{bo}$) showing that, according to Equation (12), very flexible elements in the set tend to dominate the overall stiffness, despite large stiffness values of other components.

Accurate measures of real-scale damping system stiffness are not abundant in the scientific literature. In this study, experimental values reported by Akcelyan et al. [34] (listed in Table 1) are used. These data correspond to real-scale state-of-the-art tests performed in the Tokyo Institute of Technology on damping units composed of damper, brace extender and the corresponding connection elements (clevises and brackets) in both edges. The set did not include gusset plates or frontal plate connections; addition of these elements would result in further reduction of stiffness. In the table, $k_{bb}$ has been calculated as $A_b \cdot E / L_b$, where $E$ is the young modulus of steel (taken as 200 Gpa), $k_{bo}$ has been found using Equation (10) and $k_{bb}$* has been estimated as $A_b \cdot E / L$, thus corresponding to the value typically adopted in practical analysis. The last column shows that the actual stiffness

($k_b$) varies between 13% and 28% of the stiffness $k_{bb}$* found with this usual assumption, a remarkable deviation. The very moderate values of the damper stiffness ($k_{bd}$) and auxiliary elements ($k_{bo}$) are noteworthy, as these two values dominate the total stiffness of the set $k_b$. After testing, the devices were installed in a five-storey real-scale structure subjected to further tests in the E-Defense shaking table. The direction, storey of installation of device, and fundamental period $T$ of the test structure are listed.

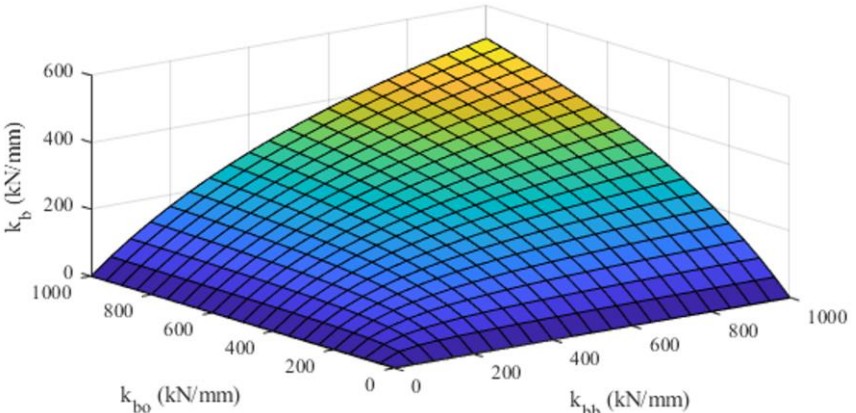

**Figure 5.** Overall stiffness $k_b$ as a function of component stiffnesses $k_{bb}$ and $k_{bo}$.

**Table 1.** Properties of non-linear viscous damping systems (after Akcelyan et al. [34]) [1].

| Direction | T | Storey | L | $L_b$ | $A_b$ [2] | $c_b$ | a | $k_{bd}$ | $k_{bb}$ | $k_{bo}$ | $k_b$ | $k_{bb}$* | $k_b/k_{bb}$* |
|---|---|---|---|---|---|---|---|---|---|---|---|---|---|
| | s | | mm | mm | mm² | kN·(s/mm)^0.38 | | kN/mm | kN/mm | kN/mm | kN/mm | kN/mm | - |
| X | 0.536 | 4–3 | 4025 | 2429 | 9121 | 49 | 0.38 | 119 | *751* | *144* | 60 | *453* | *13%* |
| | | 2 | 3947 | 2104 | 8380 | 98 | 0.38 | 193 | *797* | *315* | 104 | *425* | *24%* |
| | | 1 | 4706 | 2864 | 8380 | 98 | 0.38 | 193 | *585* | *332* | 101 | *356* | *28%* |
| Y | 0.575 | 4–3 | 3947 | 2104 | 8380 | 98 | 0.38 | 193 | *797* | *315* | 104 | *425* | *24%* |
| | | 2 | 3849 | 1542 | 15,323 | 196 | 0.38 | 438 | *1987* | *357* | 179 | *796* | *22%* |
| | | 1 | 4629 | 2322 | 15,323 | 196 | 0.38 | 438 | *1320* | *356* | 171 | *662* | *26%* |

[1] Inferred data is shown in italics. [2] Cross-section area of brace extender. For other symbols refer to main text and Figure 4.

These data have been used to find equivalent values for a longer damper unit under the assumption of similar damper forces, which leads to equal damper and mounting elements. Thus, $k_{bd}$ and $k_{bo}$ are unchanged, as is the difference between $L$ and $L_b$ representing the total length of damper and auxiliary elements. Length $L$ is replaced by $L'$ = 8006 mm, thereby finding $L_b'$ as $L' - (L - L_b)$. $k_{bb}$, $k_b$ and $k_{bb}$* are recalculated accordingly. For this case, the ratio $k_b/k_{bb}$* takes values between 23% and 40%, that is, a smaller influence than in the previous case but still very remarkable. It can be argued that, because the brace element is longer, the cross-section area should be increased due to buckling and robustness considerations. If that is the case, the ratio $k_b/k_{bb}$* decreases further, thus strengthening the conclusion that $k_b$ is actually a fraction of $k_{bb}$* somewhere between, say, 10% and 50%. These limits are, of course, loosely set, based on existing data and should be taken cautiously.

A relevant question is whether this conclusion can be applied to linear dampers. Unfortunately, to these authors' knowledge, no real-scale accurate data has been published for systems with this type of devices, probably because current seismic practice favors smaller exponents. In order to obtain a range of magnitude, the values in Table 1 have been used with certain assumptions and simplifications. First, non-linear dampers are replaced by linear dampers resulting in similar damper forces; this allows for a similar choice connection and brace extender design. The design forces are not listed in the reference, so

they are estimated indirectly using peak drift velocity values $v_0$ derived from peak drift ratios $\theta$ assuming harmonic vibration at the natural period $T$:

$$v_0 \approx \frac{2\pi\theta h}{T},$$ (13)

where $h$ is the corresponding interstorey height. Then, from Equation (1)

$$c_{b,1}v_0 = c_b v_0^a.$$ (14)

The corresponding approximate values are listed in Table 2. A second important consideration is the value of $k_{bd}$. This value is obtained using an assumption on the relaxation time $\lambda$ of the linear damper. Constantinou and Symans [23] reported values of $\lambda = 0.006$ s for dampers with $c_b = 15.45$ N·s/mm; Reinhorn et al. [24] reported $\lambda = 0.014$ s for dampers with $c_b = 201.4$ N·s/mm; Seleemah and Constantinou [25] reported $\lambda = 0.008$ s for dampers with $c_b = 17.7$ N·s/mm; these values were found under forced harmonic excitation, for reduced-scaled linear dampers with small capacity, and suggest that larger values of $c_b$ might be associated with larger values of $\lambda$. However, in the absence of data to confirm this observation, a conservative value of 0.006 s (minimum of the values reported) is adopted here. The values of $k_{bb}$, $k_{b0}$ and $k_{bb}$* presented in Table 1 are unchanged, whereas $k_b$ is computed using Equation (12). The ratios $k_b/k_{bb}$* obtained (19–42%) are slightly larger than those for non-linear dampers, but still below 50%. Changing the assumed relaxation time does not fundamentally change the results: a relaxation time 10 times shorter (0.0006 s) results in a 26–57% range; a relaxation time 10 times larger (0.06 s) results in a 6–12% range. In both cases, the main point stands valid: there is an important reduction in Maxwell stiffness due to auxiliary elements and damper contribution. A comparison of results for $k_b/k_{bb}$* between Tables 1 and 2 hint that similar reductions are to be expected for lower exponents. This point, however, needs experimental confirmation.

**Table 2.** Properties assumed for linear viscous damping systems.

| Direction | T | Storey | θ | h | $v_0$ | $f_b$ | $c_{b,1}$ | λ | $k_{bd}$ | $k_{bb}$ | $k_{bo}$ | $k_b$ | $k_{bb}$* | $k_b/k_{bb}$* |
|---|---|---|---|---|---|---|---|---|---|---|---|---|---|---|
| | s | | % | mm | mm/s | kN | kN·(s/mm) | s | kN/mm | kN/mm | kN/mm | kN/mm | kN/mm | - |
| X | 0.536 | 4 | 0.54 | 3000 | 189 | 359 | 1.90 | 0.006 | 316 | 751 | 144 | 88 | 453 | 19% |
| | | 3 | 0.56 | 3000 | 198 | 365 | 1.85 | 0.006 | 308 | 751 | 144 | 87 | 453 | 19% |
| | | 2 | 0.64 | 3000 | 225 | 768 | 3.41 | 0.006 | 568 | 797 | 315 | 161 | 425 | 38% |
| | | 1 | 0.62 | 3485 | 253 | 803 | 3.17 | 0.006 | 528 | 585 | 332 | 151 | 356 | 42% |
| Y | 0.575 | 4 | 0.65 | 3000 | 213 | 751 | 3.53 | 0.006 | 589 | 797 | 315 | 163 | 425 | 38% |
| | | 3 | 0.74 | 3000 | 242 | 789 | 3.26 | 0.006 | 544 | 797 | 315 | 159 | 425 | 38% |
| | | 2 | 0.79 | 3000 | 259 | 1618 | 6.26 | 0.006 | 1043 | 1987 | 357 | 235 | 796 | 29% |
| | | 1 | 0.76 | 3485 | 289 | 1688 | 5.84 | 0.006 | 974 | 1320 | 356 | 218 | 662 | 33% |

### 2.3. Numerical Study

A numerical study is performed in order to investigate the influence of the reduced stiffness in models including added viscous damping systems. In this study, a case is defined as a moment resisting frame (MRF) equipped with an added damping system (ADS) and subjected to earthquake excitation represented by a ground motion set (GMS) at different intensities. Peak results for significant variables obtained for the same case using different assumptions for the Maxwell system stiffness, are compared to assess the influence of this parameter in the result variability. Hereby, the study is described in detail:

#### 2.3.1. Description of Moment Resisting Frames (MRFs)

A five-storey frame (shown in Figure 6), referred to as 05D, is analyzed in this study. In the y-direction, two of these frames provide lateral strength and stiffness to a building whose floor layout is also presented. Seismic weight for each frame is 2601 kN at a typical storey and 2440 kN at roof. The frames were designed for gravity and wind loads, but taking into account capacity criteria according to Eurocode 8: at every joint the condition $\sum M_{c,Rd} \geq 1.3 \sum M_{b,Rd}$ was imposed, where $\sum M_{c,Rd}$ and $\sum M_{b,Rd}$ represent the addition of

column and beam resisting moments concurrent at the joint, respectively. Column sizing was adjusted iteratively to ensure a complete global mechanism in a pushover analysis under a 1st mode lateral force pattern. The dynamic properties are: fundamental period $T_1$ = 2.010 s, modal participation factor $\Gamma_1$ = 1.364, total weight $W_{tot}$ = 12,847 kN, first-mode effective weight $W_{eff,1}$ = 9955 kN. Estimative interstorey yield drift ratios $\theta_y$ and storey stiffnesses $k_s$ are found using the simplified procedure suggested by Akiyama [9] (Chapter 12), rendering $\theta_y^T$ = [0.699; 0.985; 0.997; 1.098; 1.232]%; $\mathbf{k_s}$ = [63.3; 30.5; 26.7; 19.1; 14.3] kN/mm. $k_s$ is used to calculate the ratio $k_s/k_d$, as indicated below.

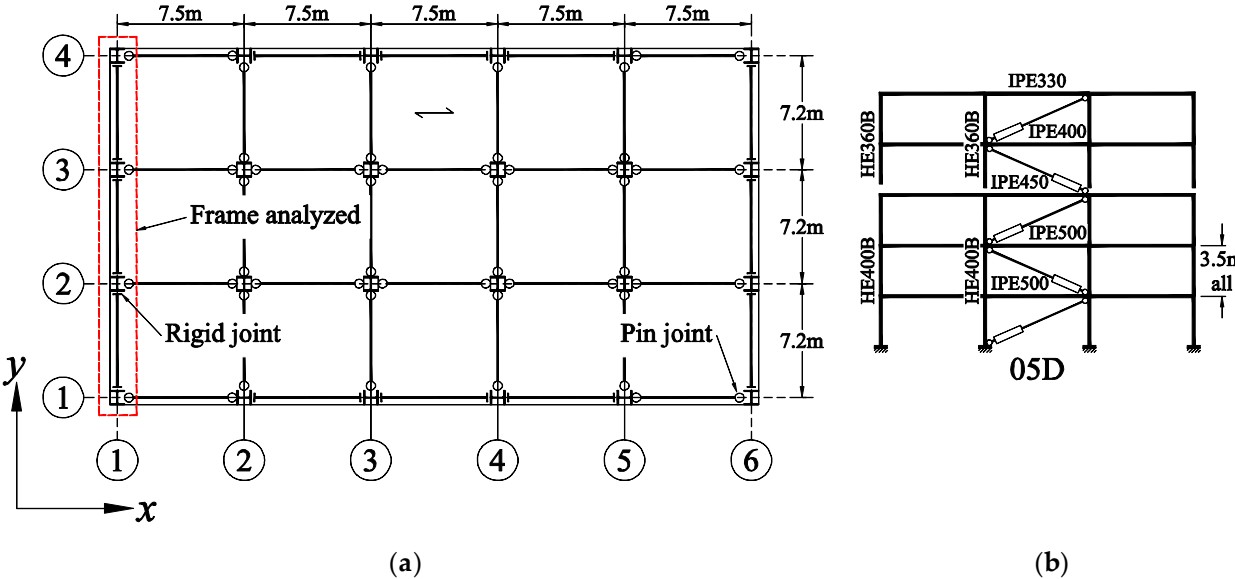

**Figure 6.** Description of building and frames used in the study: (**a**) plan layout; (**b**) definition of frame.

### 2.3.2. Description of Ground Motion Sets (GMS)

Two different GMSs have been considered: a single representative accelerogram (referred to as EUK), and a set of accelerograms (referred to as ESB). All motions have been taken from the European Strong Motion Database [40,41]. Only horizontal components are considered.

Set EUK comprises only the X-component of the Kalamata Earthquake (000414XA in the database), a far-field accelerogram recorded at ground type B (360 m/s $\leq v_{s,30}$ < 800 m/as defined in Eurocode 8) with peak horizontal acceleration *PHA* = 0.24 g, peak horizontal velocity *PHV* = 31.5 cm/s and peak horizontal displacement *PHD* = 6.57 cm, epicentral distance of 11 km, focal depth of 1 km, Arias Intensity $I_A$ = 0.553 m/s and significant duration (as defined by Bommer and Mendis [42]) $D_{5-95\%}$ = 5.13 s. The accelerogram is normalized by its PHA. The 5% elastic spectrum of this normalized accelerogram, shown in Figure A1 (Appendix A) approaches reasonably well the Eurocode 8 5% normalized elastic response spectrum throughout a wide range of periods.

Set ESB is formed by 20 non-impulsive motions, with magnitude Mw $\geq$ 5.5, epicentral distance from 10 km to 50 km, recorded at firm soil (ground type B). The motion selection is listed in Table A1, and the statistical properties of the motions are given in Table A2 (Appendix A). Both horizontal components from every record are consistently included in the set, adding up a total of 40 accelerograms. The motions are individually normalized by their PHA and pre-scaled so that the mean 5% elastic spectrum of the pre-scaled set approaches well the 5% Eurocode 8 elastic response spectrum over a wide range of periods. Spectra of pre-scaled accelerograms are shown in Figure A2 (Appendix A). TH response for set ESB is taken as the median ($\hat{x}$) of individual responses to each accelerogram in

the set, assuming a lognormal distribution; correspondingly, the 16% ($x^{16}$) and 84% ($x^{84}$) percentiles are taken as indicative of data dispersion:

$$\hat{x} = \exp(\mu(\ln x_i)), \tag{15}$$

$$x^{16} = \hat{x} / \exp(\sigma(\ln x_i)), \tag{16}$$

$$x^{84} = \hat{x} \exp(\sigma(\ln x_i)), \tag{17}$$

where, $x_i$ is the individual response to the *i*-th accelerogram, $\mu(\cdot)$ represents mean and $\sigma(\cdot)$ standard deviation. Time history analysis is extended 10 s after the end of the recorded signal, in order to achieve a state of negligible motion in the structural system.

### 2.3.3. Seismic Intensities

Intensities are defined by the value of spectral acceleration at the first mode period and 5% damping $S_a$ ($T_1$,5%), a common intensity measure hereby referred to as $S_{a1}$. Use of 5% damping allows for a direct relationship to code-defined spectral values with no intervention of damping correction factors and is consistent with the Ground Motion Set pre-scaling procedure adopted. The design intensity level for the frames was fixed at $S_{a1,D}$ = 0.22 g, corresponding to a Eurocode 8–defined 5% elastic spectrum anchored at $PHA_D$ = 0.36 g. The intensity level was fixed so that the maximum interstorey drift ratio for GMS ESB was approximately 2%. According to Eurocode 8 [43] (section 4.4.3.2), this condition implies an interstorey drift ratio of 1% (importance class II) to 0.8% (importance classes III or IV) in the damage limitation check, which for these frames happens to be a more severe check than frame strength. This situation is usual for MRFs. The inclusion of ADSs is targeted at performance enhancement, measured as reduction of interstorey drift ratio. The design intensity level is referred to with the letter 'D' (from design) and it is assumed to correspond to a return period of 475 years. Besides the design level, two additional intensity levels are defined, namely, one for which the frames with added damping remain totally elastic ($S_{a1,E}$, referred to with letter 'E', from elastic) and a rare earthquake level ($S_{a1,C}$, referred to with letter 'C', from near collapse) taken as 1.5·$S_{a1,D}$, in accordance with the relationship between design and maximum considered earthquake spectra established in ASCE/SEI 7-16 [31] (section 11.4.7). These two levels are assumed to correspond roughly to return periods of less than 95 years and about 2475 years. The analysis for GMS ESB is conducted by scaling at these three levels. For GMS EUK, the analysis is performed at 20 equally spaced intensity levels; the peak intensity was $S_{a1,max}$ = 0.55 g, corresponding to the Eurocode 8 5% elastic spectrum anchored at $PHA$ = 1.08 g.

### 2.3.4. Description of Added Damping Systems (ADS)

Added Damping Systems (ADSs) are included in order to obtain enhanced performance, measured as a reduction of interstorey drift ratio. Different damper exponents, damping ratios, and height-wise distributions of dampers are considered. An ADS in this study is described by a label of the type "Q.2.1," with the following meaning:

- The first letter ("Q" in the example) indicates the height-wise distribution (Q, R, S); the "Q" distribution has constant damping coefficients at all storeys; in the "R" and "S" distribution the damping coefficients decrease linearly with height up to a fraction of the first storey damping coefficient, 2/3 for the "R" distribution and 1/3 for the "S" distribution. Only complete vertical distributions are considered.
- The second number ("2" in the example) indicates the relative amount of damping in the first mode; for linear dampers, "1" indicates 11%, "2" indicates 23% and "3" indicates 34%; for non-linear dampers, the values are drift-dependent, but the damping coefficients keep a similar progression.
- The third number ("1" in the example) indicates the damper exponent: "1" indicates linear ($a$ = 1) and "5" indicates non-linear ($a$ = 0.5). All dampers in one ADS are either linear or non-linear.

The "Q" distribution is a realistic option, as in some cases only one type of damper is used throughout the structure. The "S" distribution represents a more efficient option, as the damper coefficients are arranged loosely following the frame storey stiffness distribution. Percentages of 11%, 23%, 34% added damping ratios in the first mode were established for ADSs Q.1.1, Q.2.1 and Q.3.1 (linear dampers); values of 20% have been recommended for optimal performance [8,44]. Non-linear damping coefficients were determined to obtain approximately similar damping ratios at $S_{a1,C}$, resulting in higher damping ratios at design level $S_{a1,D}$ (damping ratios for non-linear dampers are intensity-dependent). For ADSs type "R" and "S", the coefficients are arranged so that added damping ratio in the first mode $\xi_{a,1}$ is the same as in the corresponding ADS type "Q". Although optimized vertical distributions of damping coefficients have been proposed in the literature [14,27,29], this topic is out of the scope of this paper, as is the distribution of dampers among bays of the frame. Thus, the distributions in the study are chosen as simple as possible within realistic limits, and the position of dampers is limited to the central bay of the frame.

Brace sections are defined as European hot-finished square hollow sections (SHSH) with the following criteria: (i) S355 grade steel (characteristic yield strength $f_y$ = 355 MPa) conforming to Eurocode 3 [45] is adopted; (ii) a maximum reduced slenderness of 2 is allowed; (iii) only compact sections class 1 with a minimum wall thickness of 8 mm are considered. Buckling is assessed considering the whole centerline length. The properties of ADSs are listed in Table 3, where $k_b$ has been found considering the cross-sectional properties of the brace with the complete centerline length of the damping system. Typical values for $k_b$ suggested in literature are 175 to 525 kN/mm [8,39,44]. The values in Table 3 belong in this range and lie above $k_{s,1}$ (63.3 kN/mm), as suggested by Chen and Chai [37]. In this table, $\xi_{a,1}$ is listed at $S_{a1,D}$ for non-linear dampers.

**Table 3.** Definition of Added Damping Systems (ADSs).

| MRF | ADS | $\xi_{a,1}$ | $a$ | $c_b$ (kN·s$^a$/mm$^a$), at Storey | | | | | $k_b$ (kN/mm), at Storey | | | | |
|---|---|---|---|---|---|---|---|---|---|---|---|---|---|
| | | | | 1 | 2 | 3 | 4 | 5 | 1 | 2 | 3 | 4 | 5 |
| 05D | Q.1.1 | 0.11 | 1 | 2.33 | 2.33 | 2.33 | 2.33 | 2.33 | 167 | 198 | 167 | 167 | 174 |
| 05D | Q.2.1 | 0.23 | 1 | 4.67 | 4.67 | 4.67 | 4.67 | 4.67 | 222 | 287 | 222 | 187 | 167 |
| 05D | Q.3.1 | 0.34 | 1 | 7.00 | 7.00 | 7.00 | 7.00 | 7.00 | 237 | 237 | 287 | 222 | 167 |
| 05D | R.1.1 | 0.11 | 1 | 2.87 | 2.63 | 2.39 | 2.15 | 1.91 | 198 | 198 | 167 | 174 | 162 |
| 05D | R.2.1 | 0.23 | 1 | 5.73 | 5.26 | 4.78 | 4.30 | 3.82 | 287 | 287 | 222 | 187 | 174 |
| 05D | R.3.1 | 0.34 | 1 | 8.60 | 7.88 | 7.17 | 6.45 | 5.73 | 282 | 282 | 287 | 222 | 167 |
| 05D | S.1.1 | 0.11 | 1 | 3.72 | 3.10 | 2.48 | 1.86 | 1.24 | 187 | 187 | 167 | 147 | 112 |
| 05D | S.2.1 | 0.23 | 1 | 7.44 | 6.20 | 4.96 | 3.72 | 2.48 | 237 | 237 | 222 | 198 | 162 |
| 05D | S.3.1 | 0.34 | 1 | 11.16 | 9.30 | 7.44 | 5.58 | 3.72 | 367 | 282 | 237 | 222 | 174 |
| 05D | Q.1.5 | 0.15 | 0.5 | 30.3 | 30.3 | 30.3 | 30.3 | 30.3 | 136 | 167 | 167 | 174 | 174 |
| 05D | Q.2.5 | 0.33 | 0.5 | 60.0 | 60.0 | 60.0 | 60.0 | 60.0 | 222 | 287 | 222 | 187 | 198 |
| 05D | Q.3.5 | 0.54 | 0.5 | 89.1 | 89.1 | 89.1 | 89.1 | 89.1 | 237 | 237 | 237 | 287 | 198 |
| 05D | R.1.5 | 0.15 | 0.5 | 37.1 | 34.0 | 30.9 | 27.8 | 24.7 | 167 | 198 | 167 | 174 | 162 |
| 05D | R.2.5 | 0.33 | 0.5 | 73.4 | 67.3 | 61.2 | 55.1 | 49.0 | 287 | 287 | 222 | 187 | 167 |
| 05D | R.3.5 | 0.54 | 0.5 | 109.0 | 99.9 | 90.9 | 81.8 | 72.7 | 282 | 282 | 237 | 287 | 198 |
| 05D | S.1.5 | 0.15 | 0.5 | 47.7 | 39.8 | 31.8 | 23.9 | 15.9 | 187 | 198 | 167 | 162 | 112 |
| 05D | S.2.5 | 0.33 | 0.5 | 94.6 | 78.8 | 63.1 | 47.3 | 31.5 | 237 | 237 | 222 | 198 | 162 |
| 05D | S.3.5 | 0.54 | 0.5 | 140.5 | 117.1 | 93.6 | 70.2 | 46.8 | 367 | 282 | 237 | 222 | 167 |

### 2.3.5. Description of Analysis Models

The analysis is performed using OpenSees 3.2.1 [46,47]. The analysis is planar; no torsional effects are included. The frame is modeled with material and geometrical non-linearity. Columns and beams are modeled using force-based beam-column elements, with five integration points. Cross sections are discretized with fibers, with four elements per flange width and four elements per web height, resulting in a total of 12 midpoints. The

model includes P-M interaction but no deterioration effects, which is deemed acceptable for the moderate ductility levels reached in the systems studied, due to the beneficial effect of added viscous damping. The material properties are defined using the Giuffré-Menegotto-Pinto hysteretic model with 0.01% isotropic strain hardening ratio. The column panel zone is modeled with rigid links and an elastic spring with equivalent tangent stiffness [48]; the panel is assumed to be reinforced with cover plates to ensure elastic behavior. To include P-Δ effects, seismic vertical loads are applied on a very flexible lean column with high axial stiffness, attached to the main structure at each storey level (minus the frame tributary gravity loads, which are entered directly in the frame). Inherent damping is modeled as 2% Rayleigh damping in the first and third modes. A program-independent convergence check was performed at the end of analysis to ensure that the Energy Balance Error (EBE, as defined by Christopoulos and Filiatrault [14]) fell under 5%, otherwise reducing the analysis time step until this condition was met.

### 2.3.6. Description of Brace Stiffness Model Options

Viscous dampers are modeled using the OpenSees Viscous Damper material [49], defined for every storey $n$ by damper exponent $a_n$, damping coefficient $c_{b,n}$ and Maxwell stiffness $k_{b,n}$ ($c_{b,n}$ and $k_{b,n}$ indicate properties related to the damper axis orientation, as shown in Figure 4). For every frame and damping system, four different models are created:

- RB, modeled with a very large stiffness (1000 kN/mm).
- HB, in which the stiffness is calculated using the brace extender cross section and total diagonal centerline length.
- MB, in which the stiffness is taken as 50% that of the HB model.
- LB, in which the stiffness is taken as 25% that of the HB model.

In the first case, the Maxwell stiffness is assumed as very large, and behavior of the model is almost purely viscous; the second case corresponds to a conventional design situation in which the Maxwell stiffness is based on dependable design data; the third and fourth cases include in an approximate way the influence of the auxiliary elements and damper stiffness. As shown in previous sections, the reduction in stiffness is expected to fall somewhere between 25% and 50%; therefore, cases MB and LB correspond approximately to a lower and upper bound of the influence of the stiffness of auxiliary elements.

### 2.3.7. Variables Analyzed

The results for several relevant variables are examined using (as customary in earthquake engineering) their peak unsigned values throughout time history as representative. For energy variables the values at the end of motion are used. Because added damping systems are included as protective systems, emphasis is placed on those variables closely related to damage measure: interstorey drift ratio $\theta$ (or drift $\Delta$); residual interstorey drift ratio $\theta_r$ (or residual drift $\Delta_r$); absolute storey acceleration $a_t$; plastic strain energy at beams ($W_{beam}$), columns ($W_{column}$) or frames ($W_{frame}$); relative input energy $E_I$ [10]; energy dissipated by added damping $W_{damper}$. The following variables are also examined, but only summarized results are provided here for brevity: beam peak rotation $\theta_b$; column peak rotation $\theta_c$; structural shear $V_s$; damper shear $V_d$; total shear $V_t$; column moment $M_c$; beam moment $M_b$; column unsigned peak axial force $N_{col,min}$. Results for strain energy in structural elements have been obtained by numerical integration of generalized internal forces over generalized internal displacements. Other energy variables have also been similarly obtained by numerical integration [14]. Storey shear $V_s$ excludes damper shear $V_d$. Total shear $V_t$ includes damper shear $V_d$ and storey shear $V_s$. Due to uncoupling $V_t$ is not the addition of $V_d$ and $V_s$ for individual motions.

## 3. Results

### 3.1. Results for One Motion (GMS EUK)

Results for GMS EUK at different intensity levels are presented in Figure 7 (interstorey drift $\Delta$), Figure 8 (residual interstorey drift $\Delta_r$), Figure 9 (absolute acceleration $a_t$) and

Figure 10 (energy dissipated by dampers $W_{damper}$). The plots show the ratio between analysis results of models MB to HB (first and second columns in plots) or LB to HB (third and fourth column in plots), at every intensity level. Every storey is treated as an independent datum; for every intensity level the 50% percentile is highlighted and its value at the maximum intensity labeled. The results are binned by amount of damping at design intensity (low or high), damper type (linear or non-linear) and storey (first of roof). In the case of residual drift ratio $\Delta_r$, the results are restricted to levels for which $\Delta_r$ at HB is not negligible. Concise results for other variables are given in Appendix B, including: storey results in Figure A3 (drift velocity $\nabla$, damper shear $V_d$, storey shear $V_s$, total shear $V_t$); element results in Figure A4 (beam Moment $M_b$, beam rotation $\theta_b$, column moment $M_c$, column rotation $\theta_c$, column axial force $N_c$, storey moment $M_s$); and energy results in Figure A5 (relative energy input $E_I$, plastic strain energy dissipated by frame $W_{frame}$, column $W_{column}$ and beam $W_{beam}$). For $W_{frame}$, $W_{column}$, $W_{beam}$ only intensity levels with a non-negligible value of the variables have been selected. Even so, very low values of plastic strain are obtained for specific storeys in model type HB and, as a result, the ordinates of the ratio LB/HB and MB/HB present large peaks; the scale of the ordinate axis has been set ignoring these values.

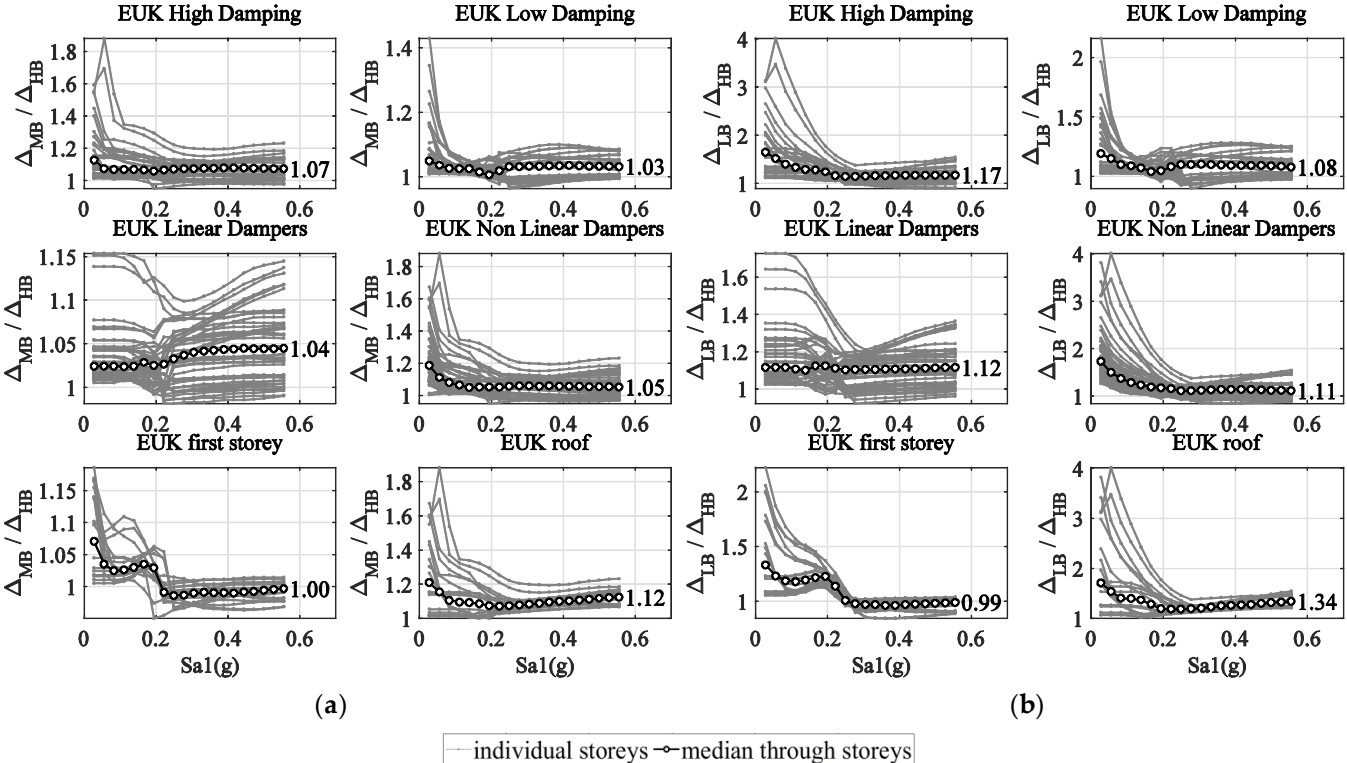

**Figure 7.** Interstorey drift $\Delta$ for frame 05D under Ground Motion Set EUK at different intensity levels, binned by damping system properties; (**a**) ratio between results for model type MB and model type HB; (**b**) ratio between results for model type LB and model type HB.

### 3.2. Results for Set of Motions (GMS ESB)

Results of the study for GMS ESB at different intensity levels are presented in Figure 11 (residual interstorey drift ratio $\theta_r$), Figure 12 (interstorey drift ratio $\theta$), Figure 13 (absolute acceleration $a_t$) and Figure 14 (plastic strain energy for frame $W_{frame}$, columns $W_{column}$, and beams $W_{beam}$). For brevity, these results are presented graphically only for ADS type "S"; results for types "Q" and "R" follow similar trends, summarized numerically in tabular form, as discussed below. At intensity level E ($S_{a1,E}$), the structure remains elastic; therefore, the values of residual drift ratio and energy dissipated by plastic strain are negligible and have been omitted. Appendix B.2 presents a summary of results for other additional

variables: relative energy input $E_I$ (Figure A6), energy dissipated by damping system $W_{damper}$ (Figure A7); plots for other relevant variables (storey shear $V_s$; damper shear $V_d$; total shear $V_t$; beam moment $M_b$; column moment $M_c$; peak column axial force $N_c$; beam rotation $\theta_b$; and column rotation $\theta_c$) are available online as Supplementary Materials.

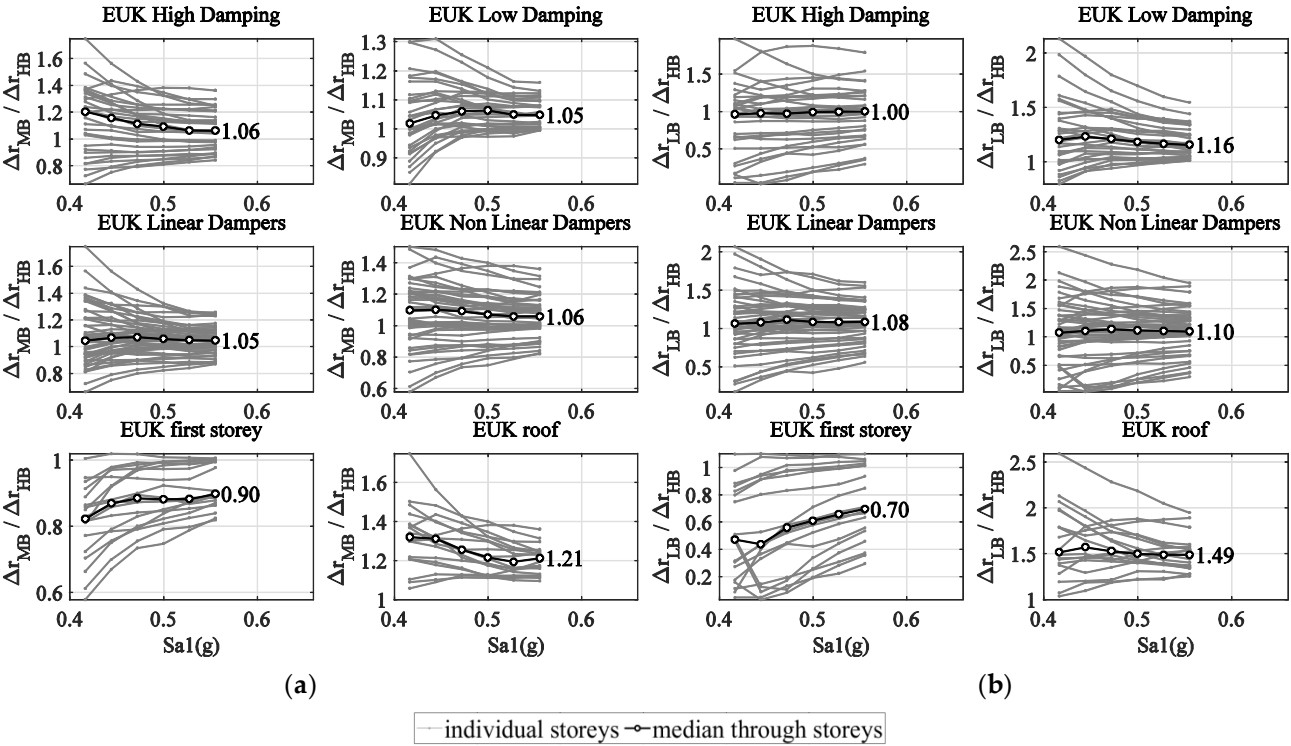

**Figure 8.** Residual interstorey drift $\Delta_r$ for frame 05D under Ground Motion Set EUK at different intensity levels, binned by damping system properties; (**a**) ratio between results for model type MB and model type HB; (**b**) ratio between results for model type LB and model type HB.

Table A3 in Appendix B.2 presents comprehensive values of ratio between median drift values $\hat{\Delta}$ obtained for models MB and LB to those obtained for model HB, at all intensity levels (E, D, C) and storeys, for all damping systems. These data are summarized in Table 4, which lists average values binned by ADS properties for the same variable ($\Delta$), thus clarifying the influence of type of damper (linear or non-linear), amount of damping (1, 2 or 3) and damper coefficient distribution (Q, R, S); an average for storeys 1 and 5 is also given, to assess the vertical variation of the ratio. The process is repeated for each variable of interest and similar summary tables are produced for: residual drift $\Delta_r$ (Table 5), absolute acceleration $a_t$ (Table 6) and plastic strain energy dissipated by frame $W_{frame}$ (Table 7). Appendix B.2 contains additional tables for other relevant variables: relative input energy $E_I$ (Table A4), energy dissipated by added damping $W_{damper}$ (Table A5), and plastic strain energy dissipated by columns $W_{column}$ (Table A6) or beams $W_{beam}$ (Table A7); additional tables for other relevant variables (storey shear $V_s$; damper shear $V_d$; total shear $V_t$; beam moment $M_b$; column moment $M_c$; column axial force $N_c$; beam rotation $\theta_b$; and column rotation $\theta_c$) are available online as Supplementary Materials.

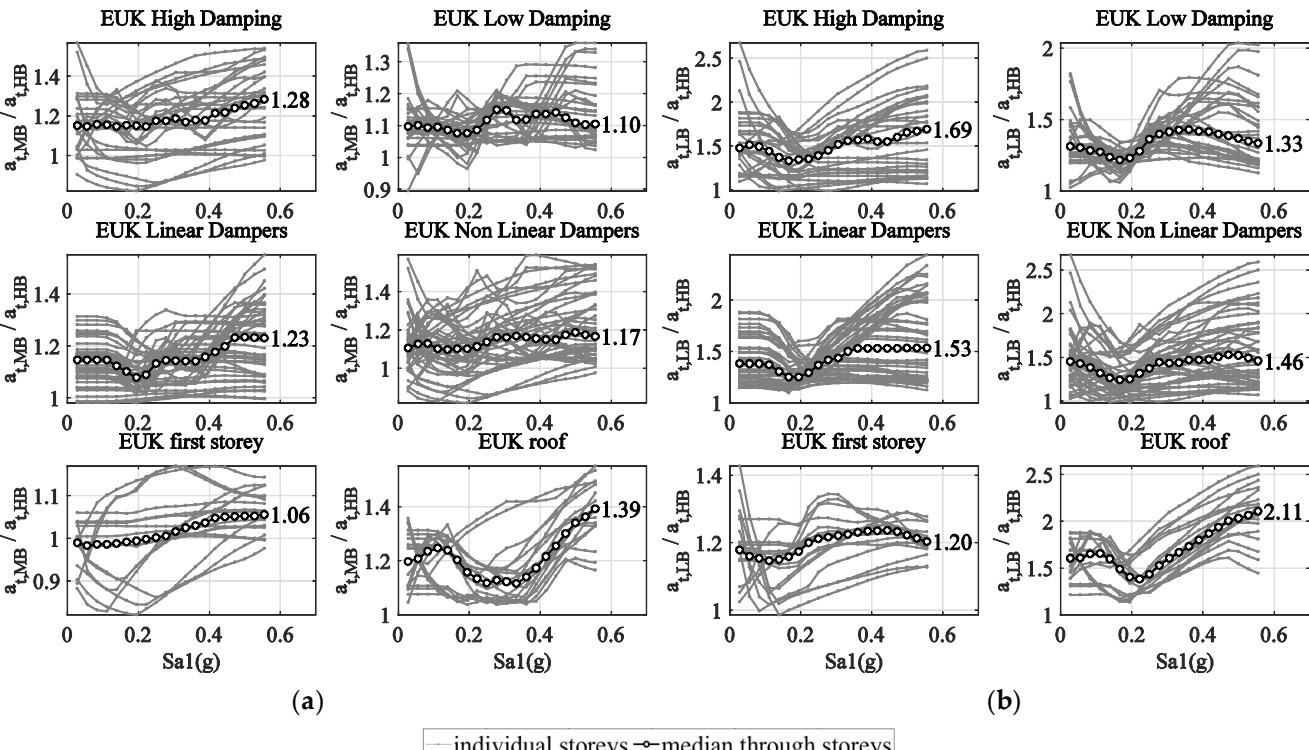

**Figure 9.** Absolute acceleration $a_t$ for frame 05D under Ground Motion Set EUK at different intensities 10. Energy dissipated by dampers $W_{damper}$ for frame 05D under Ground Motion Set EUK at different intensity levels, binned by damping system properties; (**a**) ratio between results for model type MB and model type HB; (**b**) ratio between results for model type LB and model type HB.

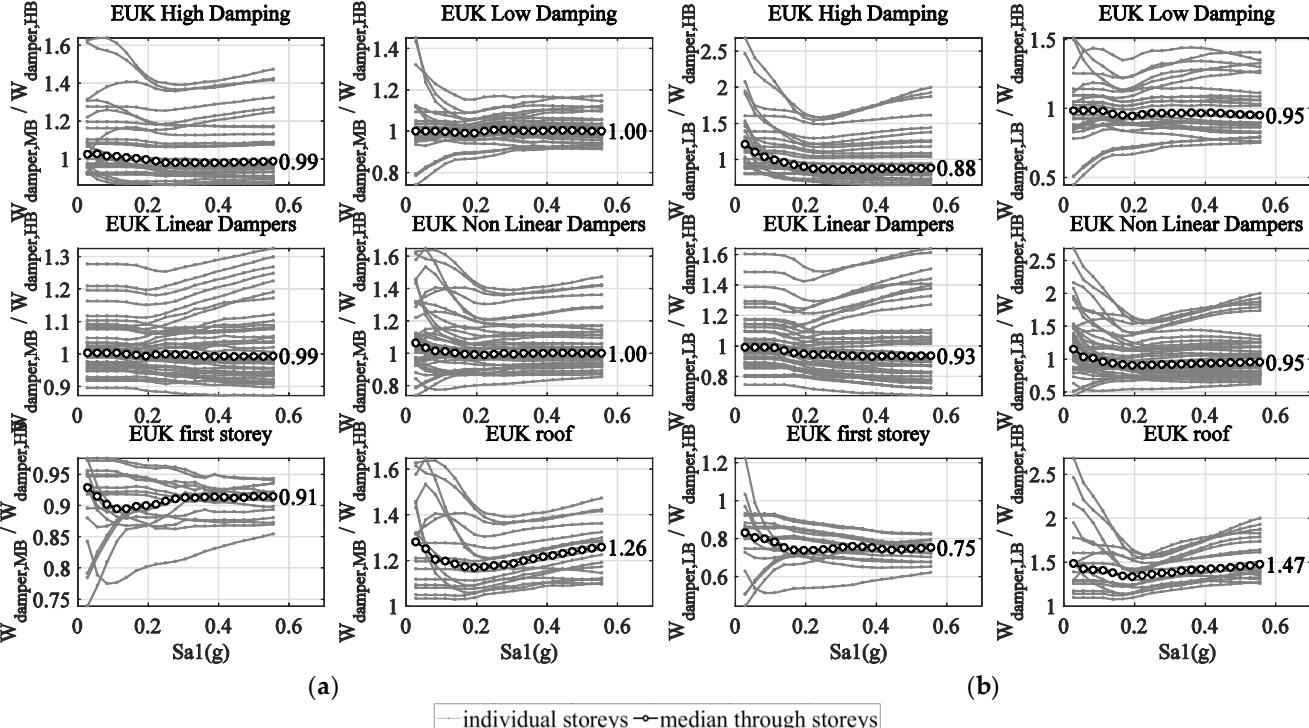

**Figure 10.** Energy dissipated by dampers $W_{damper}$ for frame 05D under Ground Motion Set EUK at different intensity levels, binned by damping system properties; (**a**) ratio between results for model type MB and model type HB; (**b**) ratio between results for model type LB and model type HB.

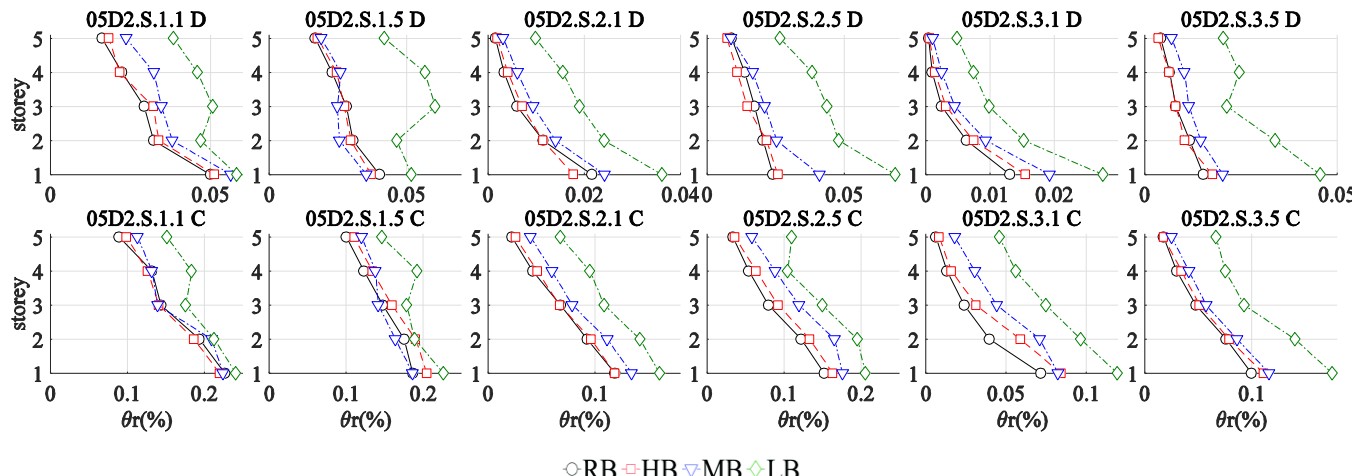

**Figure 11.** Residual drift ratio $\theta_r$ for frame 05D with Added Damping System Type S under Ground Motion Set ESB at intensity levels D ($S_{a1,D}$) and C ($S_{a1,C}$).

**Table 4.** Interstorey drift $\Delta$; summary of ratios between median and dispersion values obtained from models MB, LB and HB under Ground Motion Set ESB, binned by ADS and storey.

| ADS | | | Storey | Median $\hat{\Delta}_r$ | | | | | | | | Dispersion $\Delta^{84} - \Delta^{16}$ | | | | | | | |
| | | | | MB/HB | | | | LB/HB | | | | MB/HB | | | | LB/HB | | | |
| Type | Amount | $a$ | | E | D | C | ALL | E | D | C | ALL | E | D | C | ALL | E | D | C | ALL |
| ALL | ALL | 1 | ALL | 1.05 | 1.05 | 1.05 | 1.05 | 1.18 | 1.16 | 1.16 | 1.17 | 1.01 | 1.01 | 1.01 | 1.01 | 1.05 | 1.04 | 1.05 | 1.05 |
| ALL | ALL | 0.5 | ALL | 1.18 | 1.10 | 1.07 | 1.12 | 1.52 | 1.28 | 1.21 | 1.34 | 1.09 | 1.01 | 1.00 | 1.03 | 1.24 | 1.05 | 1.01 | 1.10 |
| ALL | 1 | ALL | ALL | 1.03 | 0.99 | 0.99 | 1.03 | 1.15 | 1.05 | 1.04 | 1.11 | 0.97 | 0.97 | 0.97 | 0.99 | 0.99 | 0.96 | 0.99 | 1.00 |
| ALL | 2 | ALL | ALL | 1.08 | 1.04 | 1.02 | 1.07 | 1.31 | 1.18 | 1.14 | 1.24 | 1.01 | 0.97 | 0.96 | 1.00 | 1.08 | 0.99 | 0.96 | 1.03 |
| ALL | 3 | ALL | ALL | 1.13 | 1.08 | 1.06 | 1.12 | 1.45 | 1.32 | 1.25 | 1.37 | 1.06 | 0.99 | 0.98 | 1.04 | 1.26 | 1.08 | 1.04 | 1.15 |
| Q | ALL | ALL | ALL | 1.13 | 1.08 | 1.05 | 1.09 | 1.39 | 1.23 | 1.17 | 1.27 | 1.05 | 1.00 | 0.97 | 1.01 | 1.16 | 1.04 | 0.96 | 1.05 |
| R | ALL | ALL | ALL | 1.11 | 1.07 | 1.06 | 1.08 | 1.33 | 1.22 | 1.19 | 1.25 | 1.04 | 1.00 | 1.01 | 1.02 | 1.14 | 1.04 | 1.06 | 1.08 |
| S | ALL | ALL | ALL | 1.11 | 1.08 | 1.06 | 1.08 | 1.32 | 1.22 | 1.18 | 1.24 | 1.05 | 1.02 | 1.03 | 1.03 | 1.14 | 1.06 | 1.08 | 1.09 |
| ALL | ALL | ALL | 1 | 1.06 | 1.03 | 1.02 | 1.04 | 1.21 | 1.15 | 1.10 | 1.15 | 1.06 | 1.05 | 1.02 | 1.04 | 1.13 | 1.11 | 1.03 | 1.09 |
| ALL | ALL | ALL | 5 | 1.26 | 1.16 | 1.13 | 1.18 | 1.71 | 1.46 | 1.41 | 1.53 | 1.05 | 0.94 | 0.96 | 0.99 | 1.19 | 0.96 | 1.00 | 1.05 |
| ALL | ALL | ALL | ALL | 1.12 | 1.07 | 1.06 | 1.08 | 1.35 | 1.22 | 1.18 | 1.25 | 1.05 | 1.01 | 1.00 | 1.02 | 1.15 | 1.04 | 1.03 | 1.07 |

**Table 5.** Residual interstorey drift $\Delta_r$; summary of ratios between median and dispersion values obtained from models MB, LB and HB under Ground Motion Set ESB binned by ADS and storey.

| | | | Storey | Median $\hat{\Delta}_r$ | | | | | | Dispersion $\Delta_r^{84} - \Delta_r^{16}$ | | | | | |
| | | | | MB/HB | | | LB/HB | | | MB/HB | | | LB/HB | | |
| Type | Amount | $a$ | | D | C | ALL | D | C | ALL | D | C | ALL | D | C | ALL |
| ALL | ALL | 1 | ALL | 1.27 | 1.18 | 1.23 | 2.41 | 1.73 | 2.07 | 1.16 | 1.12 | 1.14 | 1.79 | 1.39 | 1.59 |
| ALL | ALL | 0.5 | ALL | 1.14 | 1.10 | 1.12 | 2.29 | 1.59 | 1.94 | 1.16 | 1.10 | 1.13 | 1.62 | 1.24 | 1.43 |
| ALL | 1 | ALL | ALL | 1.04 | 1.01 | 1.03 | 1.55 | 1.20 | 1.38 | 1.04 | 1.05 | 1.05 | 1.15 | 1.03 | 1.09 |
| ALL | 2 | ALL | ALL | 1.16 | 1.14 | 1.15 | 2.30 | 1.60 | 1.95 | 1.16 | 1.08 | 1.12 | 1.52 | 1.15 | 1.34 |
| ALL | 3 | ALL | ALL | 1.30 | 1.16 | 1.23 | 2.98 | 2.01 | 2.50 | 1.17 | 1.10 | 1.14 | 2.28 | 1.63 | 1.96 |
| Q | ALL | ALL | ALL | 1.16 | 1.09 | 1.12 | 2.02 | 1.46 | 1.74 | 1.12 | 1.04 | 1.08 | 1.56 | 1.17 | 1.36 |
| R | ALL | ALL | ALL | 1.12 | 1.10 | 1.11 | 2.11 | 1.64 | 1.88 | 1.18 | 1.11 | 1.15 | 1.68 | 1.33 | 1.51 |
| S | ALL | ALL | ALL | 1.33 | 1.22 | 1.28 | 2.93 | 1.88 | 2.40 | 1.18 | 1.19 | 1.19 | 1.87 | 1.44 | 1.66 |
| ALL | ALL | ALL | 1 | 1.16 | 1.05 | 1.10 | 1.84 | 1.28 | 1.56 | 1.11 | 1.04 | 1.08 | 1.45 | 1.12 | 1.28 |
| ALL | ALL | ALL | 5 | 1.29 | 1.28 | 1.28 | 3.20 | 2.39 | 2.79 | 1.20 | 1.16 | 1.18 | 2.02 | 1.70 | 1.86 |
| ALL | ALL | ALL | ALL | 1.21 | 1.14 | 1.17 | 2.35 | 1.66 | 2.01 | 1.16 | 1.11 | 1.14 | 1.71 | 1.31 | 1.51 |



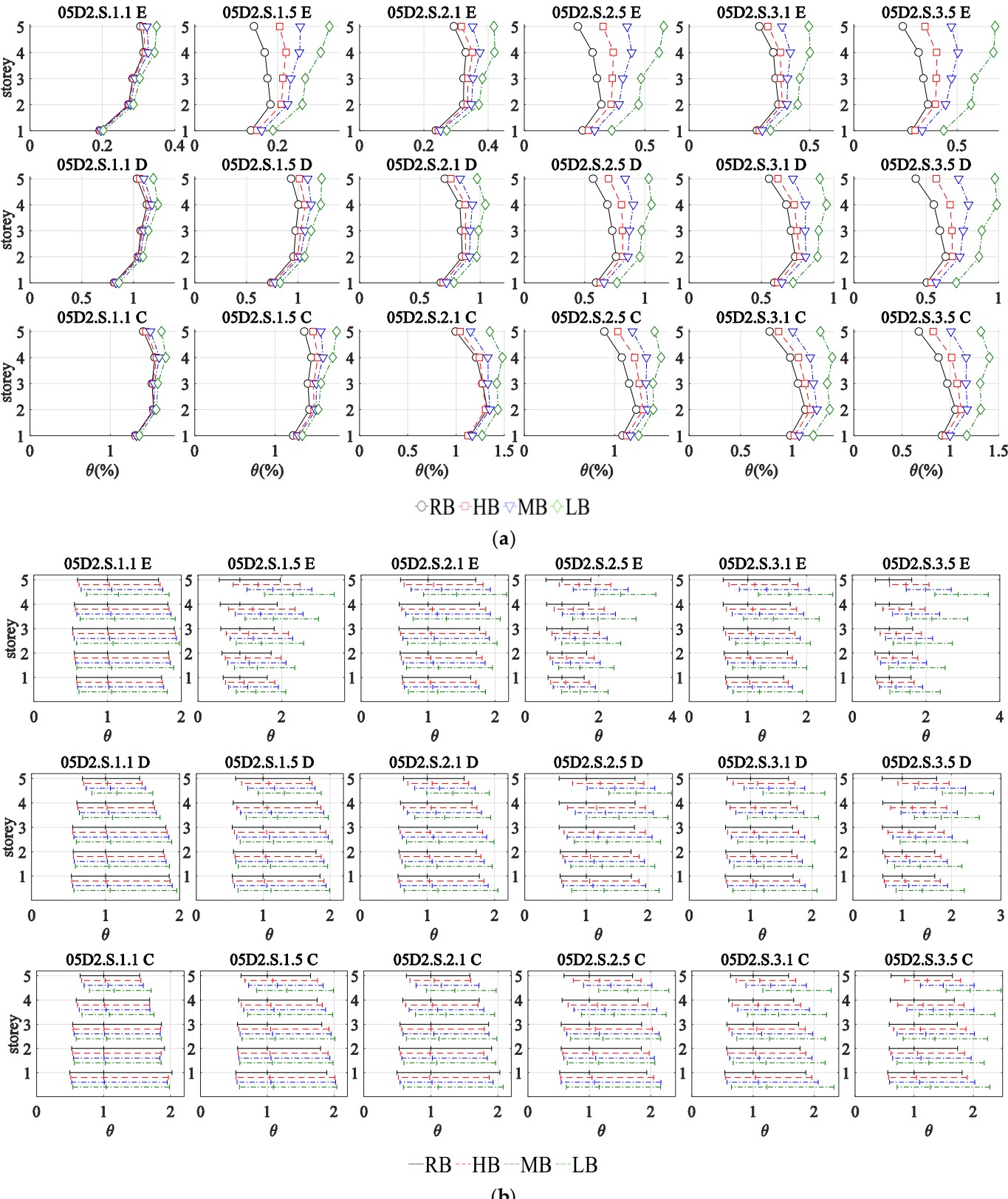

**Figure 12.** Interstorey drift ratio $\theta$ for frame 05D with Added Damping System type S, under Ground Motion Set ESB at intensity levels E ($S_{a1,E}$), D ($S_{a1,D}$) and C ($S_{a1,C}$); (**a**) absolute values for RB, HB, MB, LB models; (**b**) median ($\hat{x}$) and dispersion ($x^{16}$–$x^{84}$) values for RB, HB, MB, LB models relative to RB median.

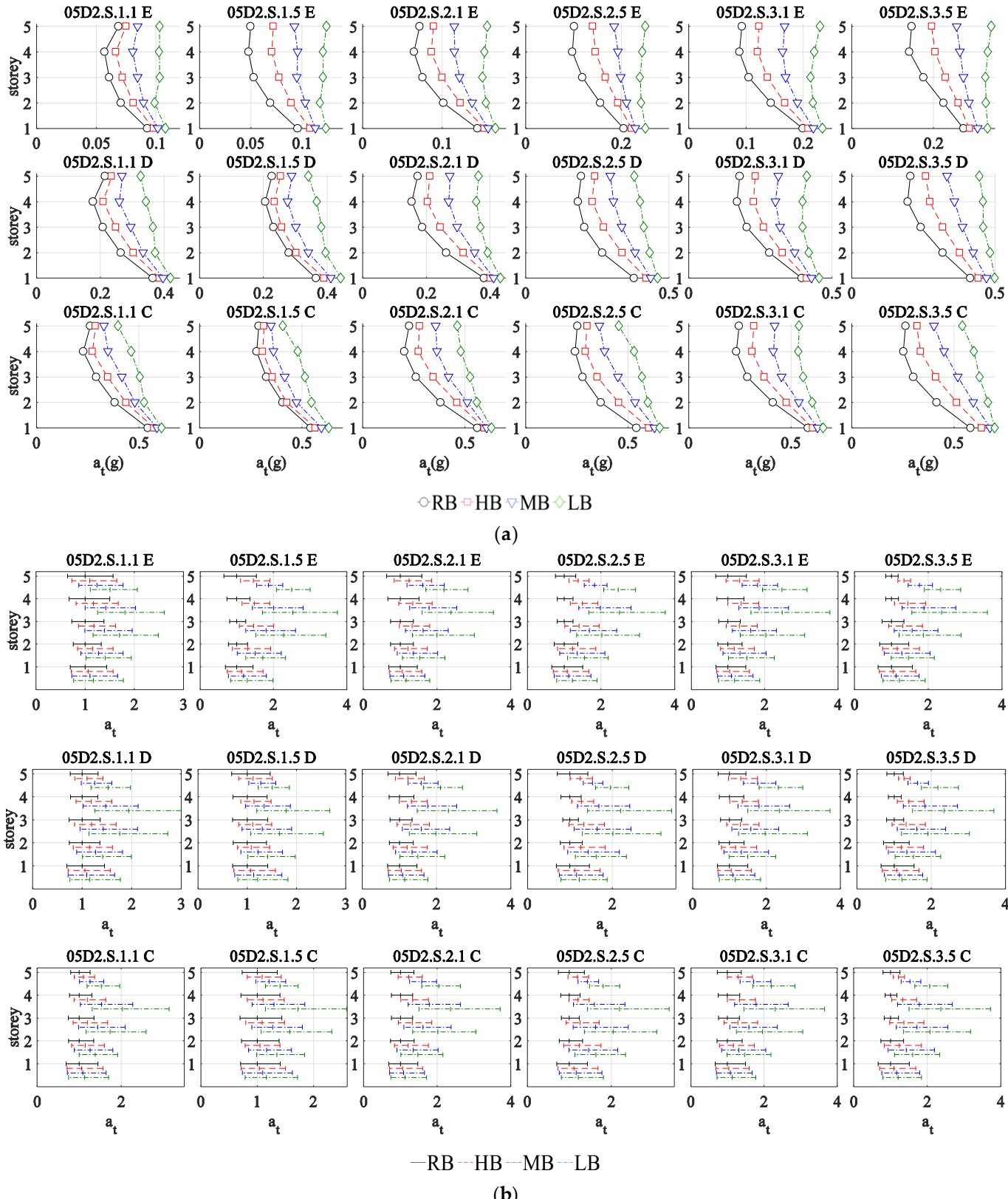

**Figure 13.** Absolute acceleration $a_t$ for frame 05D with Added Damping System type S, analyzed under Ground Motion Set ESB at intensity levels E ($S_{a1,E}$), D ($S_{a1,D}$) and C ($S_{a1,C}$); (**a**) absolute values for RB, HB, MB, LB models; (**b**) median ($\hat{x}$) and dispersion ($x^{16}$–$x^{84}$) values for RB, HB, MB, LB models relative to RB median.

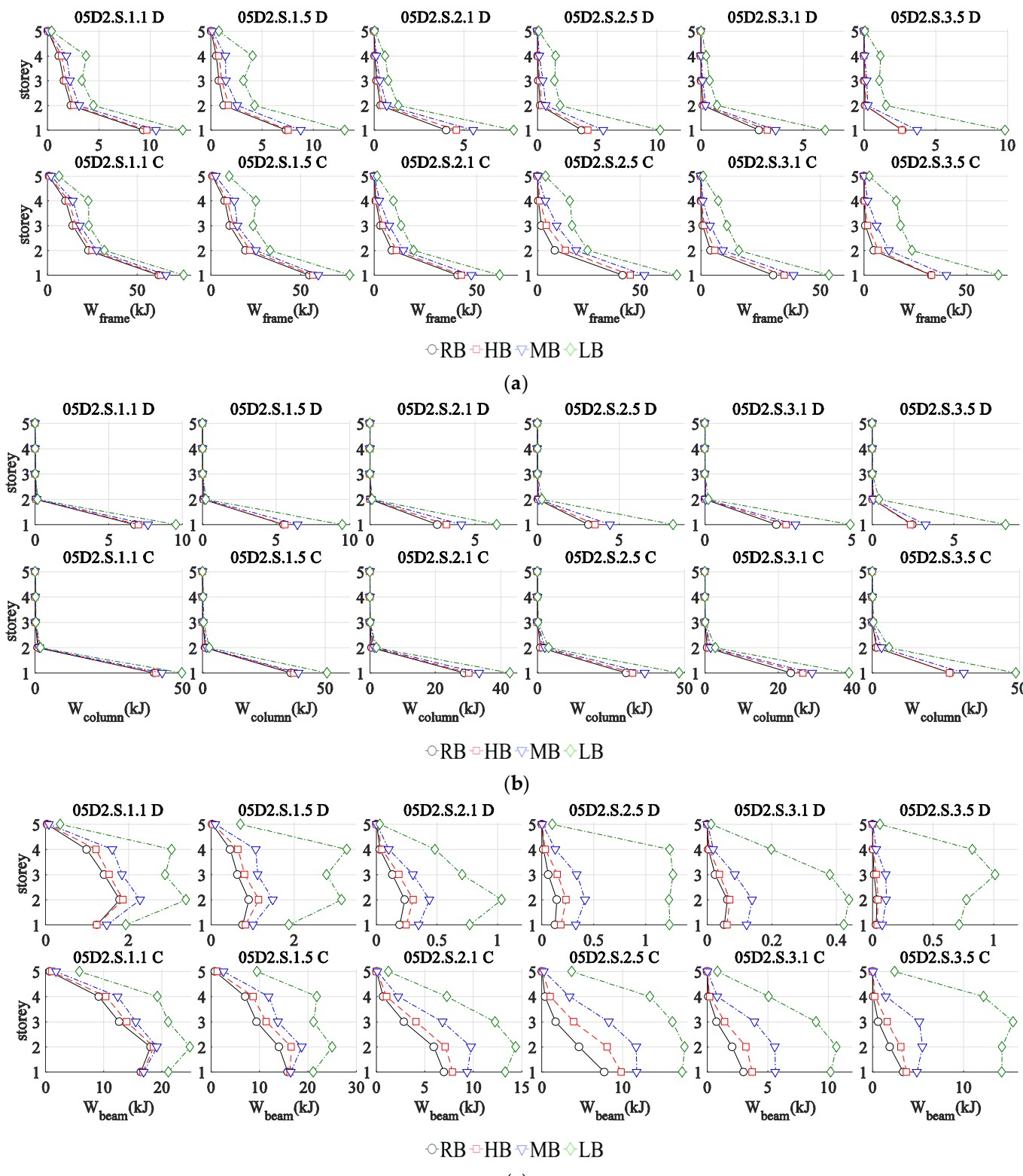

**Figure 14.** Plastic strain energy for frame 05D with Added Damping System Type S, analyzed under Ground Motion Set ESB at intensity levels D ($S_{a1,D}$) and C ($S_{a1,C}$); (**a**) total frame plastic strain energy $W_{frame}$; (**b**) column plastic strain energy $W_{column}$; (**c**) beam plastic strain energy $W_{beam}$.

**Table 6.** Absolute acceleration $a_t$; summary of ratios between median and dispersion values obtained from models MB, LB and HB under Ground Motion Set ESB binned by ADS and storey.

| ADS | | | Storey | Median $\hat{a}_t$ | | | | | | | | Dispersion $a_t^{84} - a_t^{16}$ | | | | | | | |
| | | | | MB/HB | | | | LB/HB | | | | MB/HB | | | | LB/HB | | | |
| Type | Amount | $a$ | | E | D | C | ALL | E | D | C | ALL | E | D | C | ALL | E | D | C | ALL |
|------|--------|-----|--------|------|------|------|------|------|------|------|------|------|------|------|------|------|------|------|------|
| ALL | ALL | 1 | ALL | 1.19 | 1.19 | 1.20 | 1.19 | 1.47 | 1.47 | 1.47 | 1.47 | 1.22 | 1.33 | 1.38 | 1.31 | 1.56 | 1.82 | 1.86 | 1.75 |
| ALL | ALL | 0.5 | ALL | 1.21 | 1.18 | 1.19 | 1.19 | 1.49 | 1.45 | 1.48 | 1.47 | 1.37 | 1.44 | 1.46 | 1.42 | 1.78 | 2.05 | 2.06 | 1.96 |
| ALL | 1 | ALL | ALL | 1.13 | 1.09 | 1.09 | 1.13 | 1.36 | 1.31 | 1.31 | 1.36 | 1.17 | 1.18 | 1.18 | 1.20 | 1.44 | 1.59 | 1.55 | 1.56 |
| ALL | 2 | ALL | ALL | 1.18 | 1.17 | 1.18 | 1.20 | 1.47 | 1.47 | 1.49 | 1.51 | 1.28 | 1.39 | 1.44 | 1.40 | 1.69 | 2.04 | 2.10 | 1.99 |
| ALL | 3 | ALL | ALL | 1.18 | 1.18 | 1.19 | 1.21 | 1.46 | 1.46 | 1.48 | 1.50 | 1.31 | 1.45 | 1.50 | 1.45 | 1.72 | 1.98 | 2.04 | 1.95 |
| Q | ALL | ALL | ALL | 1.21 | 1.19 | 1.20 | 1.20 | 1.49 | 1.48 | 1.51 | 1.50 | 1.30 | 1.41 | 1.44 | 1.38 | 1.68 | 1.99 | 2.05 | 1.91 |
| R | ALL | ALL | ALL | 1.20 | 1.19 | 1.19 | 1.19 | 1.48 | 1.47 | 1.48 | 1.48 | 1.29 | 1.40 | 1.44 | 1.38 | 1.69 | 1.99 | 2.04 | 1.91 |
| S | ALL | ALL | ALL | 1.20 | 1.18 | 1.19 | 1.19 | 1.46 | 1.43 | 1.43 | 1.44 | 1.28 | 1.35 | 1.38 | 1.34 | 1.64 | 1.82 | 1.79 | 1.75 |
| ALL | ALL | ALL | 1 | 1.05 | 1.05 | 1.05 | 1.05 | 1.15 | 1.12 | 1.10 | 1.12 | 1.07 | 1.08 | 1.08 | 1.07 | 1.20 | 1.19 | 1.14 | 1.18 |
| ALL | ALL | ALL | 5 | 1.28 | 1.25 | 1.25 | 1.26 | 1.69 | 1.63 | 1.63 | 1.65 | 1.18 | 1.23 | 1.33 | 1.25 | 1.48 | 1.82 | 2.00 | 1.77 |
| ALL | ALL | ALL | ALL | 1.20 | 1.19 | 1.19 | 1.19 | 1.48 | 1.46 | 1.47 | 1.47 | 1.29 | 1.38 | 1.42 | 1.37 | 1.67 | 1.93 | 1.96 | 1.85 |

**Table 7.** Plastic strain energy dissipated by frame $W_{frame}$; summary of ratios between median and dispersion values obtained from models MB, LB and HB under GMS ESB binned by ADS and storey.

| | | | Storey | Median $\hat{W}_{frame}$ | | | | | | Dispersion $W_{frame}^{84} - W_{frame}^{16}$ | | | | | |
| | | | | MB/HB | | | LB/HB | | | MB/HB | | | LB/HB | | |
| Type | Amount | $a$ | | D | C | ALL | D | C | ALL | D | C | ALL | D | C | ALL |
|------|--------|-----|--------|------|------|------|-------|-------|-------|------|------|------|-------|-------|-------|
| ALL | ALL | 1 | ALL | 1.45 | 1.59 | 1.52 | 6.57 | 10.06 | 8.32 | 1.33 | 1.36 | 1.34 | 4.90 | 5.27 | 5.09 |
| ALL | ALL | 0.5 | ALL | 1.85 | 1.94 | 1.90 | 15.13 | 16.76 | 15.95 | 1.63 | 1.47 | 1.55 | 9.19 | 7.82 | 8.51 |
| ALL | 1 | ALL | ALL | 1.30 | 1.21 | 1.25 | 3.01 | 2.40 | 2.70 | 1.14 | 1.12 | 1.13 | 1.74 | 1.54 | 1.64 |
| ALL | 2 | ALL | ALL | 1.64 | 1.67 | 1.66 | 8.56 | 9.09 | 8.83 | 1.36 | 1.35 | 1.35 | 4.69 | 4.26 | 4.47 |
| ALL | 3 | ALL | ALL | 1.86 | 2.25 | 2.06 | 19.93 | 27.45 | 23.69 | 1.78 | 1.64 | 1.71 | 14.03 | 13.20 | 13.61 |
| Q | ALL | ALL | ALL | 1.42 | 1.45 | 1.44 | 6.18 | 8.48 | 7.33 | 1.29 | 1.25 | 1.27 | 4.39 | 5.45 | 4.92 |
| R | ALL | ALL | ALL | 1.53 | 1.64 | 1.58 | 7.85 | 11.41 | 9.63 | 1.37 | 1.41 | 1.39 | 5.57 | 5.81 | 5.69 |
| S | ALL | ALL | ALL | 2.01 | 2.22 | 2.11 | 18.52 | 20.34 | 19.43 | 1.76 | 1.59 | 1.67 | 11.18 | 8.38 | 9.78 |
| ALL | ALL | ALL | 1 | 1.18 | 1.07 | 1.13 | 1.96 | 1.39 | 1.67 | 1.12 | 1.11 | 1.11 | 1.69 | 1.31 | 1.50 |
| ALL | ALL | ALL | 5 | 1.61 | 2.04 | 1.82 | 16.81 | 33.80 | 25.30 | 1.48 | 1.80 | 1.64 | 11.77 | 17.95 | 14.86 |
| ALL | ALL | ALL | ALL | 1.65 | 1.77 | 1.71 | 10.85 | 13.41 | 12.13 | 1.48 | 1.41 | 1.45 | 7.05 | 6.54 | 6.80 |

## 4. Discussion

This discussion focuses on relative change of variable results with the type of model. Values cited hereby do not refer to variable results, but to the ratio between results for two different types of model. Thus, "LB" refers to the ratio between results obtained for model type LB and those obtained for model type HB. Likewise, "MB" refers to the ratio between results for model type MB and HB. All values cited are obtained from the median and dispersion values listed on summary tables (Section 3 and Appendix B). To simplify the discussion, ratios listed in those tables under heading "MB/HB," sub-column "ALL" are referred simply as "MB." Ratios listed in those tables under heading "LB/HB," sub-column "ALL," are referred simply as "LB." The results are discussed in detail for every relevant variable hereby.

### 4.1. Drift

Variation in drift with Maxwell stiffness is moderate, on average about 1.08 (type MB) to 1.25 (type LB). Focusing on type LB, the variation in drift is:

- Larger for non-linear dampers (1.34) than for linear dampers (1.17).
- Larger for increasing damping (1.11 for low damping to 1.37 for high damping).
- Almost unaffected by the type of damper distribution (1.27, 1.25, 1.24 for distributions Q, R, S).
- Much larger for the top half of the building (1.15 for first storey, 1.53 for roof).

The influence is much larger for moderate intensity (1.35 for intensity level E, 1.18 for intensity level C). Dispersion is virtually unaffected by the type of model, except for high damping at low intensity levels with non-linear dampers (reaching up to 1.26).

### 4.2. Residual Drift

Residual drift is largely affected by the Maxwell stiffness adopted. On average, the ratio to model type HB is about 1.17 (type MB) to 2.01 (type LB). Focusing on type LB, the variation in residual drift is:

- Uninfluenced by the damper type; the ratio is very similar for linear dampers (2.07) and non-linear dampers (1.94).
- Larger for increasing damping (1.38 for low damping to 2.50 for high damping).
- Moderately influenced by the type of damper distribution (1.74; 1.88; 2.40 for distributions Q, R, S).
- Much larger for the top half of the building (1.56 for first storey, 2.79 for roof).

The influence is much larger for moderate intensity (2.35 for intensity level D vs. 1.66 for intensity level C).

Result dispersion follows similar trends to the median: dispersion is larger for LB than for MB, increases with increasing damping, is moderately affected by the distribution type, and is much larger at the roof than in the first storey.

### 4.3. Total Acceleration

Total storey acceleration is remarkably affected by the Maxwell model adopted, with an average of variation 1.19 (type MB) to 1.47 (type LB). Focusing on type LB, the variation in total acceleration is:

- Uninfluenced by the damper type (1.47 for linear dampers, 1.47 for non-linear dampers).
- Moderately influenced by the amount of damping (1,36 for low damping, 1.50 for high damping).
- Almost uninfluenced by the damper distribution (1.50 for Q, 1.44 for S).
- Larger for the roof (1.65) than for the first storey (1.15).
- Almost uninfluenced by the level of intensity (1.48 for level E vs. 1.47 for level C).

The dispersion follows similar trends, except that it increases steadily with the intensity level (1.67 at level E vs. 1.96 at level C).

### 4.4. Relative Energy Input

Relative energy input has been pointed out as a very stable quantity [9,50]; the results obtained in this study confirm that this variable is virtually unaffected by the type of model (1.01 for MB, 1.03 for LB). The largest variation for LB obtained is 1.15, in the first storey at intensity E. The change in dispersion is also very moderate, albeit slightly larger for LB (1.22) than for MB (1.17). The dispersion is slightly larger for smaller intensities and non-linear dampers.

### 4.5. Energy Dissipated by Damping System and Damper Shear

When the average throughout storeys is considered, the variations of $W_{damper}$ with Maxwell stiffness present a remarkable stability, with a slight tendency to decrease for type LB; the average values are 1.02 for MB and 0.97 for LB. These results are virtually unaffected by intensity, type of damper or type of distribution. The dispersion results show a moderate dependency on the type of damper (larger for non-linear dampers) and amount of damping (larger for higher damping).

The height-wise distribution of the variation of $W_{damper}$ is largely affected by the type of model, as shown in Figure A7 (Appendix B.2). When the Maxwell stiffness is reduced, the energy dissipated in the bottom half of the building decreases whereas the top half experiences an increase in energy dissipation. The average values for the first storey are 0.83 (MB) and 0.64 (LB). The average values for roof are 1.27 (MB) and 1.52 (LB). Similar trends are obtained for dispersion, with larger values for roof than first storey.

The observations for damper shear $V_d$ are very similar to those for $W_{damper}$, with a stable average throughout storeys close to 1, but with a reduction in first storey value (0.93 for MB, 0.83 for LB) and an increment at roof (1.12 for MB 1.23 for LB).

*4.6. Plastic Strain Energy Dissipated by Frame, Columns and Beams*

Plastic strain energy is highly influenced by the type of Maxwell model. Considering the plastic strain energy in the whole frame $W_{frame}$, the ratio increases from 1.71 (MB) to 12.13 (LB). Focusing on type LB, the variation of $W_{frame}$ is:

- Largely influenced by the type of damper (8.32 for linear dampers versus 15.95 for non-linear dampers).
- Largely influenced by the amount of damping (2.7 for low damping to 23.69 for high damping).
- Largely influenced by the damper distribution (7.33 for type Q, 19.43 for type S).
- Larger for the roof (25.30) than for the first storey (1.67).

The results are much larger for beam plastic strain $W_{beam}$ (MB 1.81, LB 27.24) than for column plastic strain $W_{column}$ (MB 1.34, LB 4.19). For columns the variation is particularly high in the bottom of the building, due to the development of plastic hinges at the column bases. The dispersion of plastic strain energy follows similar trends.

*4.7. Storey Shear and Total Shear*

Storey shear $V_s$ is clearly influenced by the Maxwell stiffness. The average variations are 1.14 (MB) and 1.40 (LB). Focusing on type LB, the trends are similar to those pointed out for drift. The variation is:

- Larger for non-linear dampers (1.50) than for linear dampers (1.31).
- Larger for increasing damping (1.19 for low damping to 1.57 for high damping).
- Almost unaffected by the type of damper distribution (1.46, 1.41, 1.34 for distributions Q, R, S).
- Much larger for the top half of the building (1.12 for first storey, 2.03 for roof).
- Larger for moderate intensity (1.52 for intensity level E, 1.33 for intensity level C).

Dispersion is larger for model type LB, particularly for high damping, non-linear dampers, distribution type Q, top storey and low intensity levels.

For total shear, the trends are similar, but the values are more moderate; this is consistent with total shear depending on both storey and damper shear.

*4.8. Member Internal Forces*

Member internal forces (column moment $M_c$ and beam moment $M_b$) are influenced by the Maxwell stiffness in a similar way. Considering $M_c$, the average variations are 1.12 (MB) and 1.34 (LB). Considering $M_b$, 1.09 (MB) and 1.24 (LB). Considering beam moment $M_b$ and models type LB, the variation is:

- Larger for non-linear dampers (1.32) than for linear dampers (1.16).
- Larger for increasing damping (1.36 for low damping to 1.09 for high damping).
- Almost unaffected by the type of damper distribution (1.28, 1.24, 1.20 for distributions Q, R, S).
- Much larger for the top half of the building (1.09 for first storey, 1.58 for roof).
- Larger for moderate intensity (1.38 for intensity level E, 1.13 for intensity level C).

Considering column moment $M_c$ and models type LB, the variation is:

- Larger for non-linear dampers (1.44) than for linear dampers (1.25).
- Larger for increasing damping (1.16 for low damping to 1.49 for high damping).
- Almost unaffected by the type of damper distribution (1.37, 1.34, 1.32 for distributions Q, R, S).
- Much larger for the top half of the building (1.09 for first storey, 1.81 for roof).
- Larger for moderate intensity (1.47 for intensity level E, 1.25 for intensity level C).

Column axial force shows little dependency on Maxwell stiffness, with ratios which are systematically close to 1, albeit slightly larger for the roof.

### 4.9. Member Rotation

Column and beam rotations are quite influenced by the Maxwell stiffness. The average variations are 1.14 (MB) and 1.45 (LB). The trends displayed are very similar to those discussed for beam and column moment.

### 4.10. Summary

The data presented proves that the value of Maxwell stiffness adopted in analysis exerts a remarkable influence on the analysis output. The results obtained for a single motion (GMS EUK) are consistent with those obtained for a set of motions (GMS ESB). Models type MB and LB show considerable variations in most of variable outputs when compared to model HB. In general, the variations are:

- Quite large for the top part of the frame and moderate or negligible at the bottom part of the frame.
- Larger for models type LB (25% Maxwell stiffness) than for models type MB (50% Maxwell stiffness).
- Larger for moderate seismic intensities than for high seismic intensities.
- Larger for high damping than for low damping.
- Larger for non-linear dampers than for linear dampers.

To obtain an overview, Table 8 lists a single value for each variable calculated averaging the values for MB and LB at all intensity levels. With the purpose of providing a practical classification of the effects of Maxwell stiffness, ratios below 1.10 are considered as "small" and left unmarked; ratios above 1.30 are considered as "large" and marked with bold characters and red font, and ratios in between are considered as "moderate" and marked with italics and blue font. The variables can then be classified as follows:

- Variables with small sensitivity to Maxwell stiffness: relative energy input, damper shear, column axial force.
- Variables with large sensitivity to Maxwell stiffness: residual drift, absolute acceleration, plastic strain energy dissipated by frame, columns and beams.

**Table 8.** Summary of variable sensitivity to Maxwell stiffness.

| Type | Amount | $a$ | Storey | $\Delta$ | $\Delta_r$ | $a_t$ | $E_I$ | $W_{damper}$ | $W_{frame}$ | $W_{column}$ | $W_{beam}$ | $V_d$ | $V_s$ | $V_t$ | $M_b$ | $M_c$ | $N_c$ | $\theta_b$ | $\theta_c$ |
|---|---|---|---|---|---|---|---|---|---|---|---|---|---|---|---|---|---|---|---|
| ALL | ALL | 1 | ALL | 1.17 | 1.59 | 1.33 | 1.02 | 1.00 | 6.92 | 3.20 | 7.54 | 1.00 | 1.27 | 1.19 | 1.16 | 1.23 | 1.02 | 1.26 | 1.29 |
| ALL | ALL | 0.5 | ALL | 1.23 | 1.53 | 1.33 | 1.03 | 0.98 | 8.92 | 3.27 | 10.09 | 1.01 | 1.34 | 1.19 | 1.22 | 1.30 | 1.02 | 1.34 | 1.37 |
| ALL | 1 | ALL | ALL | 1.07 | 1.20 | 1.24 | 1.01 | 0.97 | 1.98 | 1.93 | 2.01 | 0.98 | 1.12 | 1.13 | 1.06 | 1.11 | 1.00 | 1.11 | 1.14 |
| ALL | 2 | ALL | ALL | 1.15 | 1.55 | 1.36 | 1.01 | 0.99 | 5.24 | 2.66 | 5.40 | 0.99 | 1.26 | 1.20 | 1.15 | 1.22 | 1.01 | 1.25 | 1.28 |
| ALL | 3 | ALL | ALL | 1.24 | 1.86 | 1.35 | 1.00 | 0.99 | 12.87 | 4.70 | 14.47 | 0.99 | 1.38 | 1.21 | 1.24 | 1.32 | 1.01 | 1.38 | 1.42 |
| Q | ALL | ALL | ALL | 1.18 | 1.43 | 1.35 | 1.01 | 0.99 | 4.38 | 2.45 | 5.03 | 1.00 | 1.30 | 1.20 | 1.19 | 1.24 | 1.02 | 1.26 | 1.29 |
| R | ALL | ALL | ALL | 1.16 | 1.49 | 1.34 | 1.02 | 1.00 | 5.61 | 2.86 | 6.13 | 1.00 | 1.27 | 1.19 | 1.16 | 1.23 | 1.02 | 1.25 | 1.29 |
| S | ALL | ALL | ALL | 1.16 | 1.84 | 1.31 | 1.03 | 0.99 | 10.77 | 4.30 | 11.45 | 1.00 | 1.23 | 1.19 | 1.14 | 1.22 | 1.02 | 1.27 | 1.30 |
| ALL | ALL | ALL | 1 | 1.10 | 1.33 | 1.09 | 1.10 | 0.74 | 1.40 | 1.37 | 2.35 | 0.88 | 1.08 | 1.03 | 1.05 | 1.06 | 0.98 | 1.12 | 1.15 |
| ALL | ALL | ALL | 5 | 1.36 | 2.04 | 1.46 | 1.04 | 1.40 | 13.56 | 2.77 | 14.53 | 1.17 | 1.69 | 1.46 | 1.39 | 1.54 | 1.08 | 1.48 | 1.55 |
| ALL | ALL | ALL | ALL | 1.17 | 1.59 | 1.33 | 1.02 | 1.00 | 6.92 | 3.20 | 7.54 | 1.00 | 1.27 | 1.19 | 1.16 | 1.23 | 1.02 | 1.26 | 1.29 |

Note: Values above or equal 1.30 marked in red bold font. Values above or equal 1.10 and below 1.30 marked in blue italic font.

For all other variables the sensitivity is, in general, moderate, but:

- For non-linear dampers and/or high damping, the sensitivity of storey shear, column moment, beam and column rotation is large.
- For the top part of the frame, the sensitivity of all variables except energy input, column axial force and damper shear is large.
- Likewise, for the bottom part of the frame the sensitivity for all variables except residual drift, plastic strain energy and column rotation is small.

These results indicate that several fundamental variables for damage control (residual drift, absolute acceleration, plastic strain energy dissipated by structural members) are strongly influenced by the actual value of damping system stiffness.

### 4.11. Minimum Maxwell Stiffness

As indicated above, several authors have proposed an upper limit to the ratio $k_s/k_d$ as a mean to obtain a value of stiffness that allegedly allows the Maxwell system to be assimilated to a pure viscous damper. Limiting ratios of 0.1–0.2 have been proposed [35,36]. The data of this study are used to assess the validity of this approach. Figure 15 shows the scatter plot of eight relevant variables (drift $\Delta$, residual drift $\Delta_r$, total acceleration $a_t$, energy dissipated by damper $W_{damper}$, storey shear $V_s$, damper shear $V_d$, beam moment $M_b$, column moment $M_c$) versus $k_s/k_d$, using the analysis data for Ground Motion Set ESB at intensity level C; the $y$-axis shows the ratio of results for models type HB, MB or LB over results for model RB for each variable; limits for $k_s/k_d$ suggested in the literature have been marked with a solid ($k_s/k_d = 0.1$) and dashed ($k_s/k_d = 0.2$) lines; the value of $k_d$ used to calculate $k_s/k_d$ is the reduced value used in analysis. The plots show a poor global correlation of the ratio $k_s/k_d$ with the variability with stiffness for all variables. Many cases with high ratios $k_s/k_d$ present a variable value close to 1 (indicating an almost pure viscous behavior); conversely, cases with low ratios $k_s/k_d$ present high values of the variables (indicating large influence of the Maxwell stiffness). The condition $k_s/k_d < 0.1$ (marked in solid line) is sufficient but not necessary: when imposed, it bounds the value of the variables, but a majority of cases with small values of the variables are discarded; additionally, if the flexibility of auxiliary elements is accounted for (cases MB, LB) this limit may not be attainable even with very stiff bracing, because the overall stiffness will be dominated by the weaker element. Thus, limiting the $k_s/k_d$ ratio does not appear as a convenient condition for the practical choice of brace stiffness if the auxiliary elements are considered. From data presented above it is clear that a condition based solely on damper and storey stiffness does not capture the dependency of the variables on the Maxwell stiffness. As an example, considering drift $\Delta$ for ADS R.2.5 at design level, the ratio MB/HB is 1.04 and 1.23 for storeys 1 and 5, respectively, with $k_s/k_d$ 0.44 and 0.17; it is clear that larger $k_s/k_d$ ratios are acceptable at first storey than at roof. Thus, existing recommendations for damper stiffness based on SDOF behavior cannot be directly applied to MDOF systems. It is also apparent that the rule to determine $k_d$ simply as larger than the first storey stiffness ($k_d \geq k_{s,1}$) proposed by Chen and Chai [37] does not constitute a valid condition to determine a satisfactory Maxwell stiffness (from the point of view of damage control).

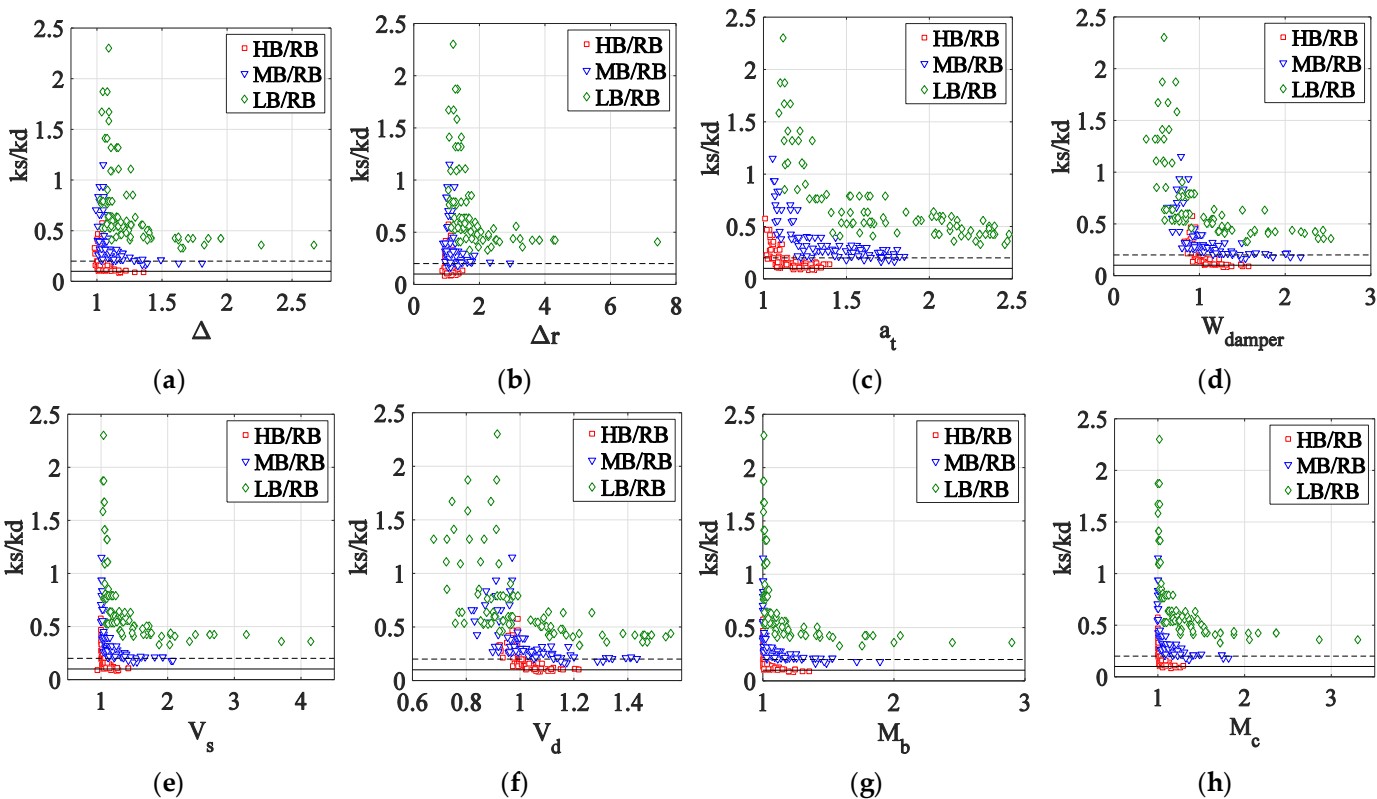

**Figure 15.** Scatter plots of ratios for several variables versus the parameter $k_s/k_d$, for Ground Motion Set ESB at intensity level C; (**a**) drift; (**b**) residual drift; (**c**) absolute acceleration; (**d**) energy dissipated by damper; (**e**) storey shear; (**f**) damper shear; (**g**) beam moment; (**h**) column moment.

## 5. Conclusions

The behavior of added damping systems is largely dependent on the flexibility of elements connecting the damper with the main structure; these include brace extenders, damping units and other auxiliary elements such as gusset plates, bolts, pins, clevises, etc. The elements are connected in series, so the overall stiffness is dominated by the most flexible element in the set. Current engineering practice neglects the influence of the auxiliary elements and favors an estimate of the system stiffness based solely on the bracing element. This is partly motivated by the limited amount of available stiffness measurements performed on complete sets.

In this work, previously published experimental data have been used to estimate that the actual stiffness of the set lies between 50% and 25% of the stiffness based purely on the bracing element. The implications of this result are studied through a numerical study on a 5-storey frame, equipped with different added damping systems, and subjected to different seismic intensity levels, in which the added damping system stiffness is consecutively scaled by 1, 0.5 and 0.25, and results for reduced and unreduced stiffness are compared. The sensitivity of the variables is defined according to their variation with damping stiffness as "small" (less than 10%), "large" (more than 30%) and "moderate" (in between). These results show that:

Considering the average throughout all storeys:

1. Relative energy input, damper shear and column axial force show a small sensitivity to the reduction in damping system stiffness.
2. Residual drift, absolute acceleration and plastic strain energy dissipated by frame elements show a large sensitivity to the reduction in damping system stiffness.

3.  For non-linear dampers and/or high damping ratios: storey shear, column moment, beam rotation and column rotation show a large sensitivity to reduction in damping system stiffness.
4.  All other variables show a moderate sensitivity to reduction in damping system stiffness.

    Considering the height-wise distribution of results:

5.  At the bottom part of the frame, only residual drift, plastic strain energy and column rotation are highly sensitive to reduction in damping system stiffness.
6.  At the top part of the frame all variables are highly sensitive to variation in damping system stiffness, except energy input, column axial force and damper shear.

The main goal of the inclusion of added viscous dampers is damage control; the study shows that three key variables related to damage (residual drift, absolute acceleration and plastic strain energy dissipated by main frame elements) are largely dependent on an accurate evaluation of the added damping system stiffness. Moreover, other relevant variables for damage control (drift, beam and column rotation, beam and column moment) also show a moderate to large dependency on this parameter. Thus, the main conclusion of this study is the need of experimental work to characterize the stiffness of added damping systems. In the absence of more accurate information at the design stage, it seems cautious to perform analyses with a reduced Maxwell stiffness based on a moderate fraction (0.25 to 0.50) of the brace extender stiffness.

As most elements in the set are currently provided by the damping unit manufacturer, it would be convenient if tests were performed on damping systems (including all auxiliary elements whose stiffness cannot be reliably estimated) and the stiffness data published as part of the damper technical information. In that way, at design stage, the engineer would choose a complete set (with clearly defined properties except for the contribution of the brace extender) instead of a damper unit (with undefined stiffness properties and unknown auxiliary elements).

The study results also show that current rules of thumb for the estimation of a minimum stiffness compatible with efficient performance of the added damping system need further refinement.

The study is subjected to the following limitations:

- Analysis was carried out for only two damper exponents ($a = 1$, $a = 0.5$). Comparison between trends for linear and non-linear cases in the study suggests an equal or larger sensitivity to Maxwell stiffness for lower damper exponents ($a < 0.5$). Additional work is needed to validate this assumption.
- The influence of horizontal distribution of dampers among bays has not been explored.
- Only simple vertical distributions of damping coefficients have been considered.

The conclusions are based on analyses performed over a flexible 5-storey moment resisting steel frame with different added damping systems subjected to ground motions recorded in firm soil. Specific results for some variables might be different for taller frames, other structural types, soft soils or incomplete vertical damper distributions. Due to scarcity of available experimental data, the results presented must be considered as preliminary.

**Supplementary Materials:** The following are available online at https://www.mdpi.com/article/ 10.3390/app11073089/s1, Figures: Figure S1 (Storey shear $V_s$ for frame 05D with Added Damping System type S, analyzed under Ground Motion Set ESB at intensity level D); Figure S2 (Damper shear $V_d$ for frame 05D with Added Damping System type S, analyzed under Ground Motion Set ESB at intensity level D); Figure S3 (Total shear $V_t$ for frame 05D with Added Damping System type S, analyzed under Ground Motion Set ESB at intensity level D); Figure S4 (Beam moment $M_b$ for frame 05D with Added Damping System type S, analyzed under Ground Motion Set ESB at intensity level D); Figure S5 (Column moment $M_c$ for frame 05D with Added Damping System type S, analyzed under Ground Motion Set ESB at intensity level D); Figure S6 (Column peak axial force $N_c$ for frame 05D with Added Damping System type S, analyzed under Ground Motion Set ESB at intensity level D); Figure S7 (Beam rotation $\theta_b$ for frame 05D with Added Damping System type S, analyzed under Ground Motion Set ESB at intensity level D); Figure S8 (Column rotation $\theta_c$ for frame 05D with

Added Damping System type S, analyzed under Ground Motion Set ESB at intensity level D). Tables: Table S1 (Storey shear $V_s$; summary of ratios between median values obtained from models MB, LB and HB under Ground Motion Set ESB binned by ADS and storey); Table S2 (Damper shear $V_d$; summary of ratios between median values obtained from models MB, LB and HB under Ground Motion Set ESB binned by ADS and storey); Table S3 (Total shear $V_t$; summary of ratios between median values obtained from models MB, LB and HB under Ground Motion Set ESB binned by ADS and storey); Table S4 (Beam moment $M_b$; summary of ratios between median values obtained from models MB, LB and HB under Ground Motion Set ESB binned by ADS and storey); Table S5 (Column moment $M_c$; summary of ratios between median values obtained from models MB, LB and HB under Ground Motion Set ESB binned by ADS and storey); Table S6 (Column peak axial force $N_c$; summary of ratios between median values obtained from models MB, LB and HB under Ground Motion Set ESB binned by ADS and storey); Table S7 (Beam rotation $\theta_b$; summary of ratios between median values obtained from models MB, LB and HB under Ground Motion Set ESB binned by ADS and storey); Table S8 (Column rotation $\theta_c$; summary of ratios between median values obtained from models MB, LB and HB under Ground Motion Set ESB binned by ADS and storey).

**Author Contributions:** Conceptualization, J.C., A.B.; methodology, J.C., A.B.; software, J.C.; validation, J.C., A.B.; writing—original draft preparation, J.C.; writing—review and editing, J.C., A.B. All authors have read and agreed to the published version of the manuscript.

**Funding:** This research received no external funding.

**Institutional Review Board Statement:** Not applicable.

**Informed Consent Statement:** Not applicable.

**Data Availability Statement:** Data available from authors on demand.

**Conflicts of Interest:** The authors declare no conflict of interest.

## Appendix A. Ground Motion Selection Information

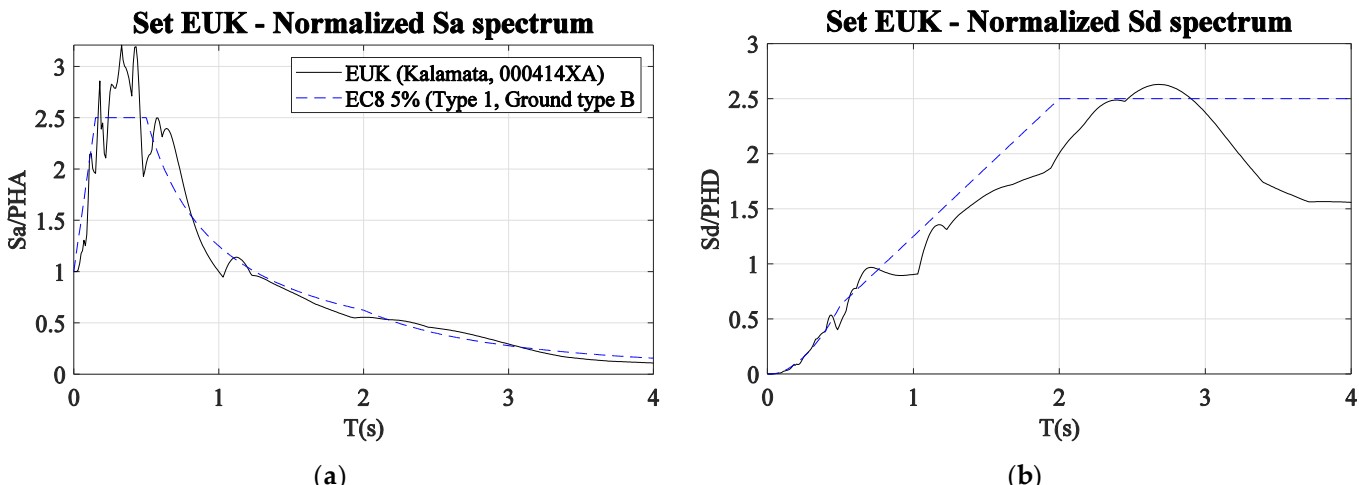

(a)  (b)

**Figure A1.** Set EUK (Kalamata, 000414XA) 5% elastic response spectrum vs. Eurocode 8 5% elastic response spectrum; (**a**) pseudo-acceleration; (**b**) displacement.

**Table A1.** Selection of accelerograms for Ground Motion Set ESB.

| Name | Station | Date | Fault | Mw | Epic. Dist. (km) | PHAX (g) | PHAY (g) | Prescale$\lambda_x$ [1] | Prescale$\lambda_y$ [1] |
|---|---|---|---|---|---|---|---|---|---|
| Friuli (aftershock) | Forgaria-Cornio | 15.09.1976 | thrust | 6 | 17 | 0.264 | 0.218 | 2.861 | 2.085 |
| Friuli (aftershock) | Forgaria-Cornio | 15.09.1976 | thrust | 6 | 17 | 0.346 | 0.336 | 2.230 | 1.537 |
| Montenegro | Bar-SkupstinaOpstine | 15.04.1979 | thrust | 6.9 | 16 | 0.375 | 0.363 | 0.909 | 0.720 |
| Montenegro (aftershock) | Tivat-Aerodrom | 24.05.1979 | thrust | 6.2 | 21 | 0.166 | 0.133 | 2.481 | 2.198 |
| Montenegro (aftershock) | Petrovac-Hotel Oliva | 15.05.1979 | oblique | 5.8 | 24 | 0.099 | 0.089 | 1.451 | 1.211 |
| Campano Lucano | Calitri | 23.11.1980 | normal | 6.9 | 16 | 0.156 | 0.176 | 0.707 | 0.768 |
| Kalamata | Kalamata-Prefecture | 13.09.1986 | normal | 5.9 | 10 | 0.215 | 0.297 | 0.819 | 0.982 |
| Kyllini | Zakynthos-OTE Building | 16.10.1988 | strike slip | 5.9 | 14 | 0.151 | 0.146 | 0.685 | 1.367 |
| Erzincan | Erzincan-Meteorologij | 13.03.1992 | strike slip | 6.6 | 13 | 0.389 | 0.513 | 0.765 | 1.164 |
| Tithorea | Aigio-OTE Building | 18.11.1992 | normal | 5.9 | 25 | 0.038 | 0.028 | 0.807 | 1.932 |
| Umbria Marche | Gubbio-Piana | 26.09.1997 | normal | 6 | 38 | 0.091 | 0.097 | 0.838 | 0.760 |
| Potenza | Brienza | 05.05.1990 | strike slip | 5.8 | 28 | 0.096 | 0.080 | 2.208 | 1.534 |
| AnoLlosia | Athens 2 (Chalandri District) | 07.09.1999 | normal | 6 | 20 | 0.110 | 0.161 | 2.121 | 2.411 |
| Griva | Edessa-Prefecture | 21.12.1990 | normal | 6.1 | 36 | 0.101 | 0.096 | 0.899 | 1.100 |
| South Aegean | Heraklio-Technical University | 23.05.1994 | oblique | 6.1 | 45 | 0.061 | 0.041 | 1.212 | 0.814 |
| Strofades | Zakynthos-OTE Building | 18.11.1997 | oblique | 6.6 | 38 | 0.131 | 0.116 | 1.152 | 1.542 |
| Kozani | Kastoria-OTE Building | 13.05.1995 | normal | 6.5 | 50 | 0.019 | 0.020 | 1.024 | 1.272 |
| Aigion | Patra-San Dimitrios Church | 15.06.1995 | normal | 6.5 | 43 | 0.084 | 0.093 | 0.713 | 0.976 |
| Duzce 1 | LDEO Station No C1058 BV | 12.11.1999 | oblique | 7.2 | 11 | 0.111 | 0.073 | 1.257 | 1.044 |
| Firuzabad | Firoozabad | 20.06.1994 | strike slip | 5.9 | 22 | 0.250 | 0.278 | 3.366 | 3.481 |

[1] Prescale factor.

**Table A2.** Statistical properties of Ground Motion Set ESB.

| | Vs30 | PHA | PHV | PHD | Epicentral Distance | Mw | Arias Intensity $I_a$ | Significant Duration $D_{5-95\%}$ |
|---|---|---|---|---|---|---|---|---|
| | m/s | g | mm/s | mm | km | | m/s | s |
| min | 365.0 | 0.019 | 11.5 | 3.4 | 10.0 | 5.80 | 0.01 | 2.77 |
| max | 800.0 | 0.513 | 1017.7 | 276.0 | 50.0 | 7.20 | 3.02 | 47.61 |
| mean | 505.4 | 0.165 | 168.0 | 39.1 | 24.9 | 6.24 | 0.46 | 16.05 |
| median | 488.0 | 0.124 | 93.2 | 15.0 | 21.5 | 6.05 | 0.22 | 10.41 |
| std | 104.4 | 0.118 | 200.9 | 53.5 | 12.3 | 0.41 | 0.65 | 12.17 |

Note: properties obtained before scaling.

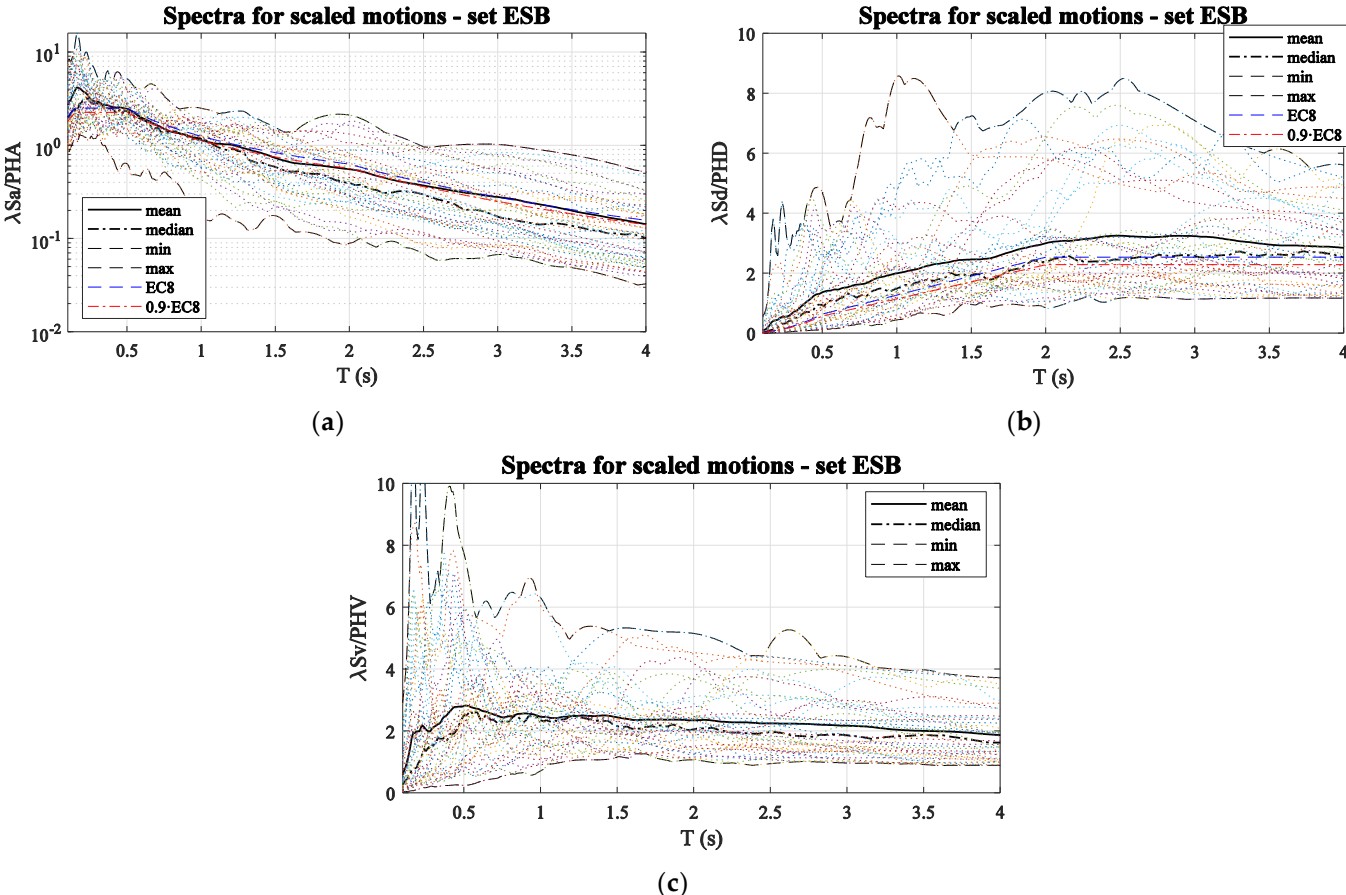

(a)

(b)

(c)

**Figure A2.** Ground motion set ESB: 5% elastic response spectrum vs. Eurocode 8 5% elastic response spectrum; (**a**) pseudo-acceleration; (**b**) displacement; (**c**) pseudo-velocity (Note: Eurocode 8 spectrum for pseudo-velocity is undefined).

# Appendix B. Additional Results

*Appendix B.1. Additional Results for Set EUK*

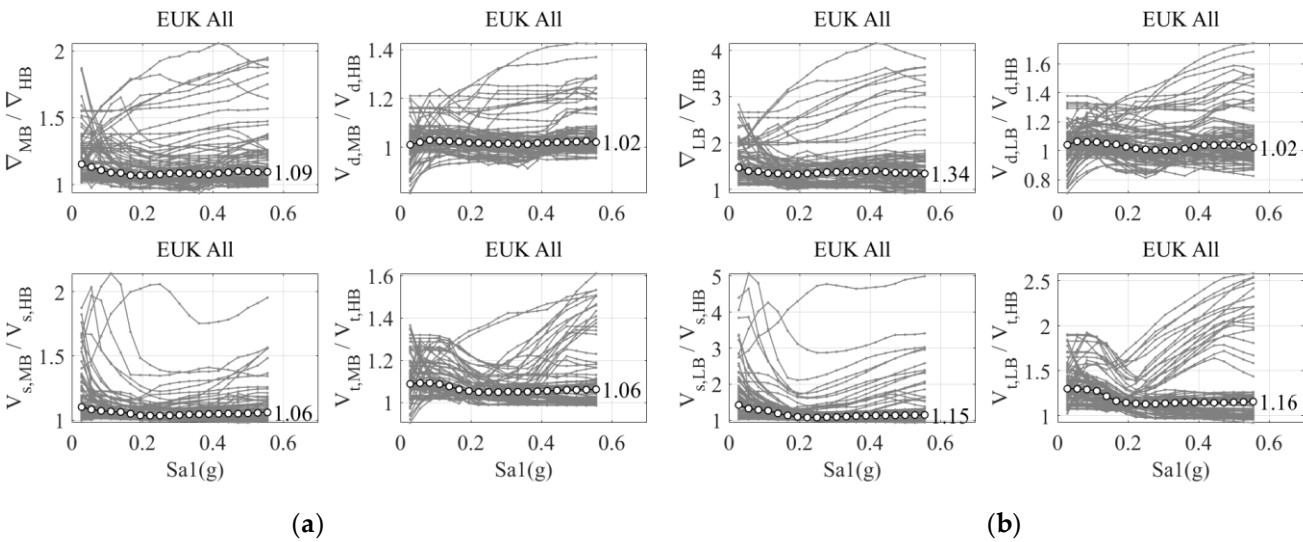

(**a**)                                              (**b**)

**Figure A3.** Summary of results for storeys (drift velocity $\nabla$ damper shear $V_d$; storey shear $V_s$; total Scheme 05D) under Ground Motion Set EUK at different intensity levels, binned by damping system properties; (**a**) ratio between results for model type MB and model type HB; (**b**) ratio between results for model type LB and model type HB.

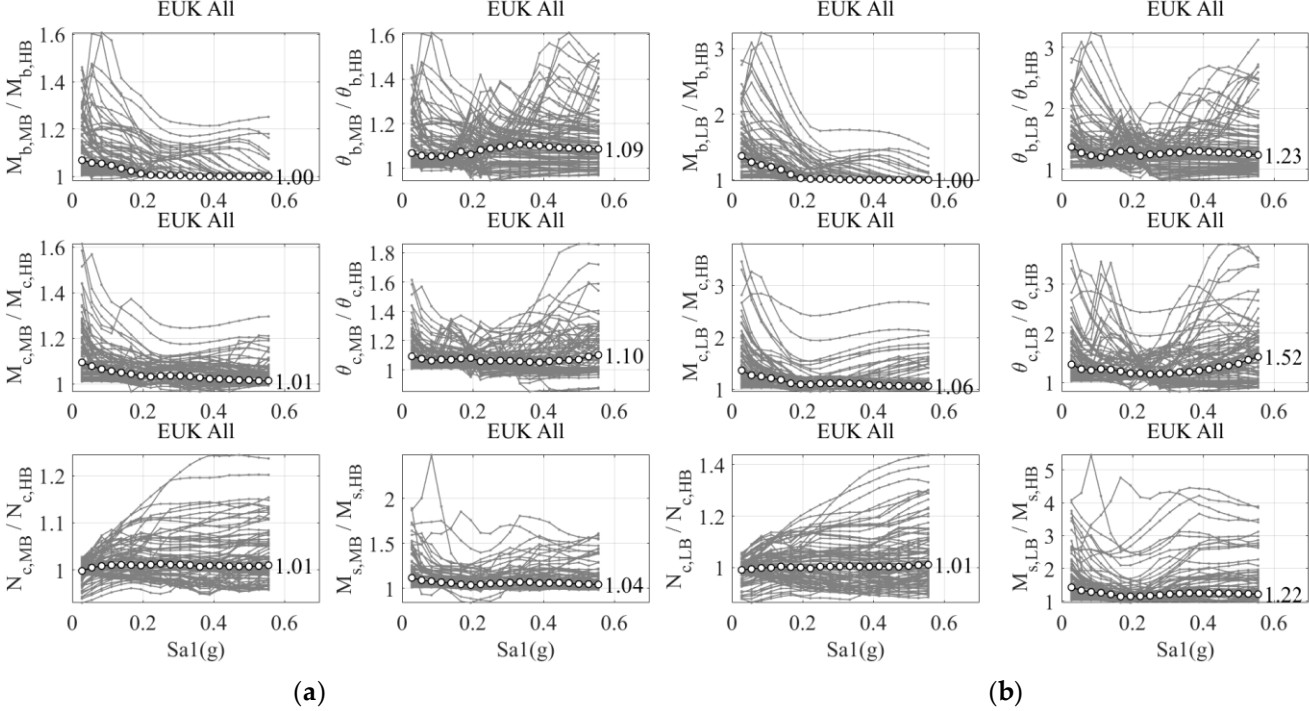

(**a**)                                              (**b**)

**Figure A4.** Summary of results for structural elements (beam moment $M_b$; beam rotation $\theta_b$; column moment $M_c$; column rotation $\theta_c$; column axial force $N_c$; storey moment $M_s$) for frame 05D under Ground Motion Set EUK at different intensity levels, binned by damping system properties; (**a**) ratio between results for model type MB and model type HB; (**b**) ratio between results for model type LB and model type HB.

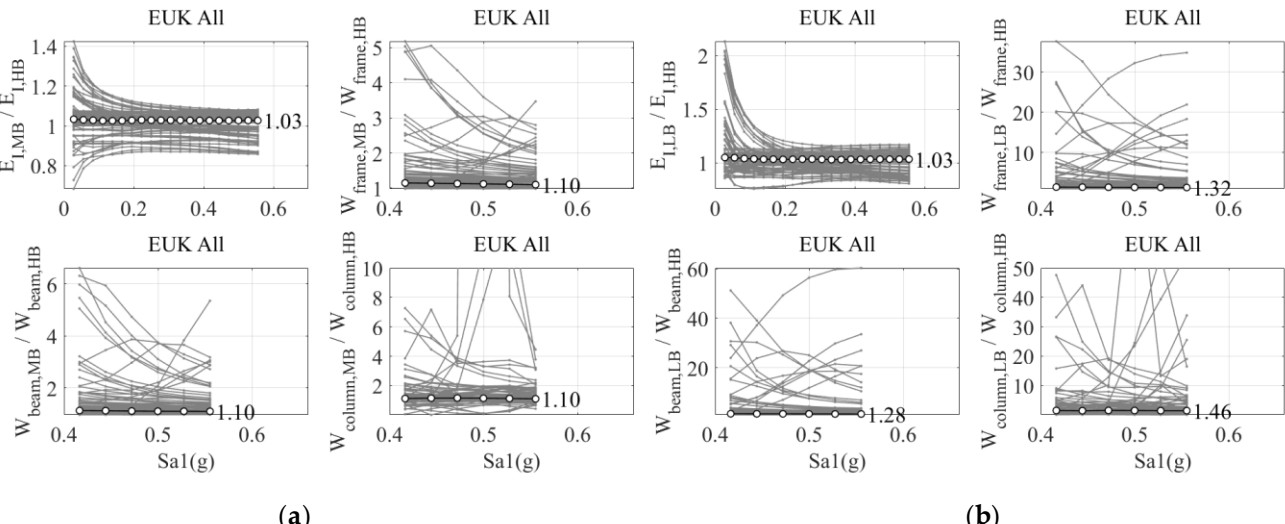

**Figure A5.** Summary of results for energy variables (relative input energy $E_I$; plastic strain energy dissipated by frame $W_{frame}$, beams $W_{beam}$ and columns $W_{column}$) for frame 05D under Ground Motion Set EUK at different intensity levels, binned by damping system properties; (**a**) ratio between results for model type MB and model type HB; (**b**) ratio between results for model type LB and model type HB.

*Appendix B.2. Additional Results for Set ESB*

**Table A3.** Interstorey drift $\Delta$; ratio between median values obtained from models MB and LB to median values obtained from model HB, $\hat{\Delta}_{MB}/\hat{\Delta}_{HB}$, $\hat{\Delta}_{LB}/\hat{\Delta}_{HB}$.

| Intensity | | E ($S_{a1,E}$) | | | | | D ($S_{a1,D}$) | | | | | C ($S_{a1,C}$) | | | | |
|---|---|---|---|---|---|---|---|---|---|---|---|---|---|---|---|---|
| Storey | | 1 | 2 | 3 | 4 | 5 | 1 | 2 | 3 | 4 | 5 | 1 | 2 | 3 | 4 | 5 |
| MB/HB | 05D2.Q.1.1 | 1.01 | 1.01 | 1.01 | 1.02 | 1.02 | 1.01 | 1.01 | 1.01 | 1.02 | 1.02 | 1.01 | 1.01 | 1.01 | 1.02 | 1.02 |
| | 05D2.Q.2.1 | 1.02 | 1.02 | 1.03 | 1.06 | 1.12 | 1.03 | 1.02 | 1.03 | 1.06 | 1.10 | 1.03 | 1.02 | 1.03 | 1.06 | 1.10 |
| | 05D2.Q.3.1 | 1.04 | 1.04 | 1.06 | 1.12 | 1.25 | 1.04 | 1.04 | 1.06 | 1.11 | 1.23 | 0.99 | 0.99 | 1.02 | 1.09 | 1.20 |
| | 05D2.R.1.1 | 1.01 | 1.01 | 1.01 | 1.02 | 1.02 | 1.01 | 1.01 | 1.01 | 1.02 | 1.02 | 1.01 | 1.01 | 1.01 | 1.02 | 1.03 |
| | 05D2.R.2.1 | 1.02 | 1.02 | 1.03 | 1.06 | 1.10 | 1.02 | 1.02 | 1.03 | 1.06 | 1.10 | 1.02 | 1.02 | 1.03 | 1.06 | 1.09 |
| | 05D2.R.3.1 | 1.04 | 1.04 | 1.06 | 1.11 | 1.22 | 1.04 | 1.04 | 1.06 | 1.11 | 1.20 | 1.05 | 1.04 | 1.06 | 1.11 | 1.19 |
| | 05D2.S.1.1 | 1.01 | 1.01 | 1.02 | 1.02 | 1.04 | 1.02 | 1.01 | 1.02 | 1.02 | 1.04 | 1.01 | 1.01 | 1.01 | 1.02 | 1.04 |
| | 05D2.S.2.1 | 1.03 | 1.04 | 1.04 | 1.07 | 1.11 | 1.04 | 1.04 | 1.04 | 1.06 | 1.10 | 1.04 | 1.03 | 1.04 | 1.07 | 1.10 |
| | 05D2.S.3.1 | 1.04 | 1.05 | 1.07 | 1.11 | 1.18 | 1.05 | 1.05 | 1.07 | 1.10 | 1.17 | 1.05 | 1.05 | 1.07 | 1.11 | 1.16 |
| | 05D2.Q.1.5 | 1.06 | 1.05 | 1.07 | 1.17 | 1.32 | 1.02 | 1.02 | 1.03 | 1.05 | 1.08 | 1.01 | 1.01 | 1.02 | 1.04 | 1.06 |
| | 05D2.Q.2.5 | 1.09 | 1.06 | 1.13 | 1.28 | 1.54 | 1.04 | 1.03 | 1.05 | 1.13 | 1.25 | 0.96 | 0.96 | 1.01 | 1.08 | 1.18 |
| | 05D2.Q.3.5 | 1.11 | 1.10 | 1.19 | 1.32 | 1.63 | 1.05 | 1.04 | 1.10 | 1.20 | 1.42 | 1.04 | 1.04 | 1.07 | 1.17 | 1.33 |
| | 05D2.R.1.5 | 1.04 | 1.04 | 1.06 | 1.11 | 1.20 | 1.02 | 1.02 | 1.03 | 1.05 | 1.07 | 1.01 | 1.01 | 1.02 | 1.04 | 1.06 |
| | 05D2.R.2.5 | 1.07 | 1.05 | 1.11 | 1.22 | 1.40 | 1.04 | 1.03 | 1.06 | 1.13 | 1.23 | 1.02 | 1.03 | 1.05 | 1.10 | 1.17 |
| | 05D2.R.3.5 | 1.12 | 1.11 | 1.20 | 1.30 | 1.54 | 1.05 | 1.04 | 1.11 | 1.19 | 1.35 | 1.03 | 1.04 | 1.08 | 1.15 | 1.26 |
| | 05D2.S.1.5 | 1.08 | 1.07 | 1.09 | 1.15 | 1.24 | 1.03 | 1.02 | 1.04 | 1.06 | 1.08 | 1.02 | 1.01 | 1.03 | 1.04 | 1.06 |
| | 05D2.S.2.5 | 1.10 | 1.09 | 1.12 | 1.21 | 1.31 | 1.05 | 1.05 | 1.07 | 1.12 | 1.19 | 1.04 | 1.04 | 1.06 | 1.11 | 1.16 |
| | 05D2.S.3.5 | 1.11 | 1.13 | 1.18 | 1.26 | 1.37 | 1.07 | 1.07 | 1.12 | 1.19 | 1.27 | 1.05 | 1.06 | 1.09 | 1.15 | 1.22 |
| LB/HB | 05D2.Q.1.1 | 1.03 | 1.03 | 1.04 | 1.06 | 1.10 | 1.03 | 1.03 | 1.03 | 1.06 | 1.09 | 1.04 | 1.03 | 1.03 | 1.06 | 1.10 |
| | 05D2.Q.2.1 | 1.09 | 1.08 | 1.10 | 1.20 | 1.40 | 1.10 | 1.07 | 1.09 | 1.19 | 1.36 | 1.04 | 1.02 | 1.05 | 1.17 | 1.34 |
| | 05D2.Q.3.1 | 1.15 | 1.14 | 1.16 | 1.37 | 1.78 | 1.17 | 1.13 | 1.15 | 1.33 | 1.69 | 1.11 | 1.07 | 1.11 | 1.31 | 1.66 |
| | 05D2.R.1.1 | 1.03 | 1.03 | 1.04 | 1.06 | 1.10 | 1.04 | 1.03 | 1.04 | 1.06 | 1.09 | 1.08 | 1.07 | 1.07 | 1.09 | 1.13 |
| | 05D2.R.2.1 | 1.09 | 1.08 | 1.10 | 1.19 | 1.35 | 1.09 | 1.08 | 1.10 | 1.18 | 1.32 | 1.09 | 1.07 | 1.09 | 1.19 | 1.32 |
| | 05D2.R.3.1 | 1.15 | 1.14 | 1.17 | 1.35 | 1.68 | 1.17 | 1.13 | 1.16 | 1.32 | 1.60 | 1.13 | 1.09 | 1.12 | 1.29 | 1.57 |
| | 05D2.S.1.1 | 1.05 | 1.04 | 1.06 | 1.08 | 1.12 | 1.05 | 1.04 | 1.06 | 1.07 | 1.12 | 1.04 | 1.02 | 1.04 | 1.07 | 1.13 |
| | 05D2.S.2.1 | 1.11 | 1.11 | 1.13 | 1.20 | 1.32 | 1.13 | 1.10 | 1.13 | 1.19 | 1.29 | 1.13 | 1.10 | 1.12 | 1.19 | 1.29 |
| | 05D2.S.3.1 | 1.16 | 1.17 | 1.21 | 1.32 | 1.52 | 1.17 | 1.16 | 1.20 | 1.31 | 1.47 | 1.18 | 1.16 | 1.19 | 1.31 | 1.46 |
| | 05D2.Q.1.5 | 1.21 | 1.19 | 1.21 | 1.43 | 1.82 | 1.08 | 1.07 | 1.07 | 1.14 | 1.21 | 1.05 | 1.04 | 1.04 | 1.10 | 1.18 |
| | 05D2.Q.2.5 | 1.34 | 1.28 | 1.36 | 1.76 | 2.54 | 1.18 | 1.11 | 1.15 | 1.34 | 1.68 | 1.04 | 1.01 | 1.09 | 1.27 | 1.55 |
| | 05D2.Q.3.5 | 1.42 | 1.38 | 1.48 | 1.87 | 2.71 | 1.29 | 1.20 | 1.26 | 1.53 | 2.18 | 1.11 | 1.05 | 1.14 | 1.41 | 1.96 |
| | 05D2.R.1.5 | 1.15 | 1.14 | 1.16 | 1.29 | 1.51 | 1.08 | 1.06 | 1.08 | 1.14 | 1.19 | 1.04 | 1.04 | 1.05 | 1.11 | 1.18 |
| | 05D2.R.2.5 | 1.30 | 1.25 | 1.29 | 1.60 | 2.10 | 1.19 | 1.13 | 1.17 | 1.33 | 1.61 | 1.12 | 1.10 | 1.13 | 1.28 | 1.49 |
| | 05D2.R.3.5 | 1.45 | 1.41 | 1.49 | 1.82 | 2.48 | 1.28 | 1.21 | 1.27 | 1.50 | 1.94 | 1.20 | 1.14 | 1.21 | 1.41 | 1.75 |
| | 05D2.S.1.5 | 1.26 | 1.24 | 1.25 | 1.38 | 1.59 | 1.09 | 1.07 | 1.10 | 1.14 | 1.21 | 1.05 | 1.04 | 1.06 | 1.12 | 1.20 |
| | 05D2.S.2.5 | 1.36 | 1.32 | 1.32 | 1.51 | 1.77 | 1.22 | 1.17 | 1.19 | 1.31 | 1.47 | 1.12 | 1.09 | 1.12 | 1.24 | 1.39 |
| | 05D2.S.3.5 | 1.47 | 1.44 | 1.46 | 1.69 | 1.98 | 1.33 | 1.27 | 1.30 | 1.48 | 1.71 | 1.23 | 1.18 | 1.22 | 1.39 | 1.60 |

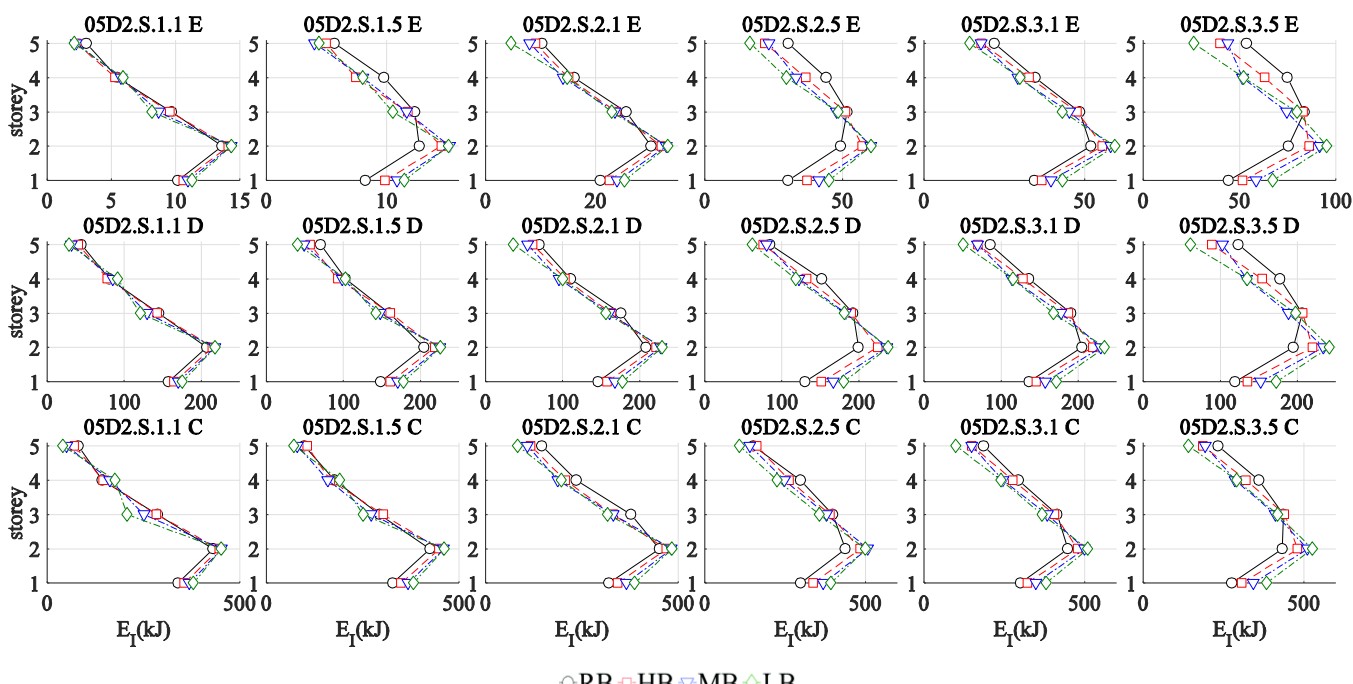

**Figure A6.** Relative energy input for frame 05D with Added Damping System Type S, analyzed under Ground Motion Set ESB at intensity levels E ($S_{a1,E}$), D ($S_{a1,D}$) and C ($S_{a1,C}$).

**Table A4.** Relative energy input $E_I$; summary of ratios between median and dispersion values obtained from models MB, LB and HB under Ground Motion Set ESB binned by ADS and storey.

| ADS | | | Storey | Median $\hat{E}_I$ | | | | | | | | Dispersion $E_I^{84} - E_I^{16}$ | | | | | | | |
|---|---|---|---|---|---|---|---|---|---|---|---|---|---|---|---|---|---|---|---|
| | | | | MB/HB | | | | LB/HB | | | | MB/HB | | | | LB/HB | | | |
| Type | Amount | $a$ | | E | D | C | ALL | E | D | C | ALL | E | D | C | ALL | E | D | C | ALL |
| ALL | ALL | 1 | ALL | 1.01 | 1.00 | 0.99 | 1.00 | 1.05 | 1.03 | 1.02 | 1.03 | 1.13 | 1.14 | 1.14 | 1.14 | 1.20 | 1.20 | 1.19 | 1.20 |
| ALL | ALL | 0.5 | ALL | 1.02 | 1.01 | 1.01 | 1.01 | 1.05 | 1.04 | 1.02 | 1.04 | 1.22 | 1.20 | 1.17 | 1.20 | 1.29 | 1.26 | 1.19 | 1.25 |
| ALL | 1 | ALL | ALL | 1.00 | 0.96 | 0.95 | 0.99 | 1.04 | 1.00 | 1.00 | 1.04 | 1.08 | 1.10 | 1.12 | 1.12 | 1.13 | 1.15 | 1.16 | 1.17 |
| ALL | 2 | ALL | ALL | 0.98 | 0.98 | 0.99 | 1.00 | 1.00 | 1.01 | 1.00 | 1.02 | 1.14 | 1.11 | 1.07 | 1.13 | 1.20 | 1.17 | 1.12 | 1.19 |
| ALL | 3 | ALL | ALL | 0.97 | 0.98 | 0.97 | 0.99 | 1.00 | 1.00 | 0.97 | 1.01 | 1.19 | 1.19 | 1.17 | 1.21 | 1.28 | 1.25 | 1.17 | 1.26 |
| Q | ALL | ALL | ALL | 1.01 | 1.00 | 0.99 | 1.00 | 1.04 | 1.04 | 1.00 | 1.03 | 1.16 | 1.16 | 1.13 | 1.15 | 1.22 | 1.19 | 1.13 | 1.18 |
| R | ALL | ALL | ALL | 1.01 | 0.99 | 1.00 | 1.00 | 1.06 | 1.04 | 1.04 | 1.05 | 1.18 | 1.20 | 1.17 | 1.18 | 1.23 | 1.22 | 1.20 | 1.22 |
| S | ALL | ALL | ALL | 1.03 | 1.03 | 1.01 | 1.03 | 1.04 | 1.03 | 1.02 | 1.03 | 1.18 | 1.16 | 1.17 | 1.17 | 1.29 | 1.27 | 1.24 | 1.27 |
| ALL | ALL | ALL | 1 | 1.07 | 1.06 | 1.06 | 1.06 | 1.15 | 1.13 | 1.12 | 1.13 | 1.11 | 1.08 | 1.07 | 1.09 | 1.14 | 1.11 | 1.09 | 1.12 |
| ALL | ALL | ALL | 5 | 1.08 | 1.05 | 1.03 | 1.05 | 1.05 | 1.03 | 1.00 | 1.03 | 1.08 | 1.08 | 1.08 | 1.08 | 1.16 | 1.17 | 1.09 | 1.14 |
| ALL | ALL | ALL | ALL | 1.02 | 1.01 | 1.00 | 1.01 | 1.05 | 1.04 | 1.02 | 1.03 | 1.17 | 1.17 | 1.16 | 1.17 | 1.24 | 1.23 | 1.19 | 1.22 |

**Table A5.** Energy dissipated by added damping $W_{damper}$; summary of ratios between median and dispersion values obtained from models MB, LB and HB under Ground Motion Set ESB binned by ADS and storey.

| ADS | | | Storey | Median $\hat{W}_{damper}$ | | | | | | | | Dispersion $W_{damper}^{84} - W_{damper}^{16}$ | | | | | | | |
|---|---|---|---|---|---|---|---|---|---|---|---|---|---|---|---|---|---|---|---|
| | | | | MB/HB | | | | LB/HB | | | | MB/HB | | | | LB/HB | | | |
| Type | Amount | $a$ | | E | D | C | ALL | E | D | C | ALL | E | D | C | ALL | E | D | C | ALL |
| ALL | ALL | 1 | ALL | 1.03 | 1.02 | 1.01 | 1.02 | 1.02 | 0.99 | 0.97 | 0.99 | 1.02 | 1.01 | 1.01 | 1.01 | 1.04 | 1.02 | 1.01 | 1.02 |
| ALL | ALL | 0.5 | ALL | 1.01 | 1.02 | 1.02 | 1.01 | 0.95 | 0.96 | 0.95 | 0.95 | 1.17 | 1.09 | 1.07 | 1.11 | 1.25 | 1.13 | 1.08 | 1.16 |
| ALL | 1 | ALL | ALL | 0.97 | 0.97 | 0.97 | 0.99 | 0.93 | 0.93 | 0.93 | 0.95 | 1.00 | 0.98 | 0.97 | 1.00 | 1.02 | 0.98 | 0.97 | 1.01 |
| ALL | 2 | ALL | ALL | 0.99 | 0.99 | 0.99 | 1.01 | 0.96 | 0.96 | 0.95 | 0.98 | 1.05 | 1.01 | 1.00 | 1.05 | 1.11 | 1.04 | 1.01 | 1.08 |
| ALL | 3 | ALL | ALL | 0.99 | 0.99 | 0.99 | 1.01 | 0.96 | 0.95 | 0.92 | 0.97 | 1.11 | 1.07 | 1.05 | 1.10 | 1.19 | 1.11 | 1.05 | 1.14 |
| Q | ALL | ALL | ALL | 1.02 | 1.02 | 1.01 | 1.02 | 0.99 | 0.98 | 0.95 | 0.97 | 1.09 | 1.05 | 1.02 | 1.05 | 1.14 | 1.06 | 1.01 | 1.07 |
| R | ALL | ALL | ALL | 1.02 | 1.02 | 1.02 | 1.02 | 0.99 | 0.99 | 0.98 | 0.99 | 1.09 | 1.05 | 1.04 | 1.06 | 1.14 | 1.08 | 1.06 | 1.09 |
| S | ALL | ALL | ALL | 1.01 | 1.02 | 1.02 | 1.01 | 0.96 | 0.97 | 0.96 | 0.96 | 1.09 | 1.07 | 1.06 | 1.07 | 1.15 | 1.10 | 1.06 | 1.10 |
| ALL | ALL | ALL | 1 | 0.82 | 0.84 | 0.84 | 0.83 | 0.63 | 0.65 | 0.64 | 0.64 | 0.97 | 0.97 | 0.96 | 0.97 | 0.86 | 0.87 | 0.83 | 0.86 |
| ALL | ALL | ALL | 5 | 1.29 | 1.27 | 1.26 | 1.27 | 1.56 | 1.51 | 1.50 | 1.52 | 1.24 | 1.17 | 1.17 | 1.19 | 1.50 | 1.39 | 1.40 | 1.43 |
| ALL | ALL | ALL | ALL | 1.02 | 1.02 | 1.02 | 1.02 | 0.98 | 0.98 | 0.96 | 0.97 | 1.09 | 1.05 | 1.04 | 1.06 | 1.14 | 1.08 | 1.04 | 1.09 |

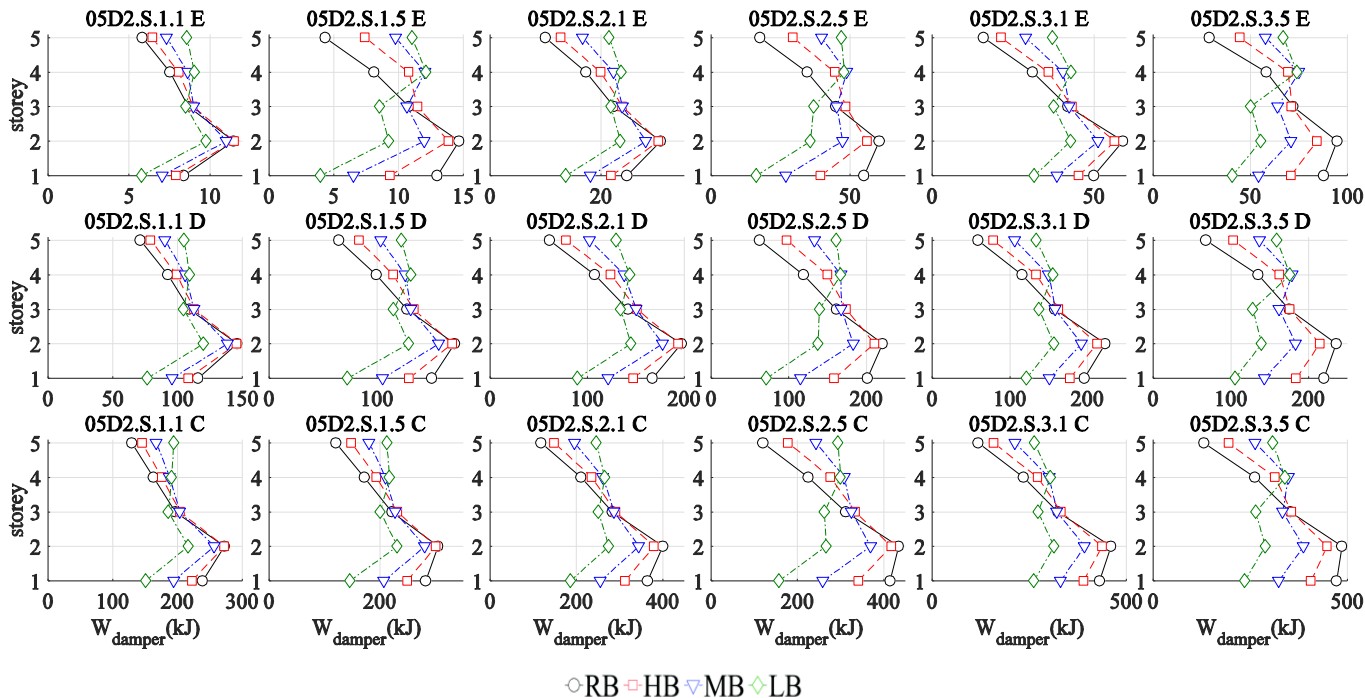

**Figure A7.** Energy dissipated by damping system $W_{damper}$ for frame 05D with Added Damping System Type S at levels D ($S_{a1,D}$) and C ($S_{a1,C}$); absolute values for RB, HB, MB, LB models.

**Table A6.** Plastic strain energy dissipated by columns $W_{column}$; summary of ratios between median values obtained from models MB, LB and HB under Ground Motion Set ESB binned by ADS and storey.

| | | | | Median $\hat{W}_{column}$ | | | | | | Dispersion $W_{column}^{84} - W_{column}^{16}$ | | | | | |
| | | | Storey | MB/HB | | | LB/HB | | | MB/HB | | | LB/HB | | |
| Type | Amount | a | | D | C | ALL | D | C | ALL | D | C | ALL | D | C | ALL |
|---|---|---|---|---|---|---|---|---|---|---|---|---|---|---|---|
| ALL | ALL | 1 | ALL | 1.27 | 1.55 | 1.41 | 4.31 | 5.38 | 4.85 | 1.25 | 1.33 | 1.29 | 3.09 | 3.96 | 3.53 |
| ALL | ALL | 0.5 | ALL | 1.42 | 1.54 | 1.48 | 3.92 | 6.21 | 5.06 | 1.28 | 1.29 | 1.29 | 3.00 | 4.02 | 3.51 |
| ALL | 1 | ALL | ALL | 1.26 | 1.26 | 1.26 | 2.57 | 2.65 | 2.61 | 1.07 | 1.10 | 1.09 | 1.51 | 1.94 | 1.72 |
| ALL | 2 | ALL | ALL | 1.26 | 1.39 | 1.33 | 3.53 | 4.45 | 3.99 | 1.20 | 1.22 | 1.21 | 2.49 | 3.04 | 2.77 |
| ALL | 3 | ALL | ALL | 1.39 | 1.84 | 1.61 | 5.85 | 9.73 | 7.79 | 1.41 | 1.48 | 1.45 | 4.84 | 6.60 | 5.72 |
| Q | ALL | ALL | ALL | 1.29 | 1.34 | 1.32 | 3.28 | 3.87 | 3.57 | 1.16 | 1.13 | 1.14 | 2.21 | 2.55 | 2.38 |
| R | ALL | ALL | ALL | 1.27 | 1.49 | 1.38 | 3.53 | 5.14 | 4.34 | 1.22 | 1.27 | 1.25 | 2.62 | 3.27 | 2.94 |
| S | ALL | ALL | ALL | 1.47 | 1.81 | 1.64 | 5.53 | 8.39 | 6.96 | 1.42 | 1.54 | 1.48 | 4.31 | 6.15 | 5.23 |
| ALL | ALL | ALL | 1 | 1.17 | 1.06 | 1.11 | 1.88 | 1.37 | 1.63 | 1.11 | 1.12 | 1.11 | 1.63 | 1.31 | 1.47 |
| ALL | ALL | ALL | 5 | 1.35 | 1.33 | 1.34 | 3.67 | 4.71 | 4.19 | 1.25 | 1.17 | 1.21 | 3.21 | 3.53 | 3.37 |
| ALL | ALL | ALL | ALL | 1.35 | 1.55 | 1.45 | 4.11 | 5.80 | 4.96 | 1.27 | 1.31 | 1.29 | 3.05 | 3.99 | 3.52 |

**Table A7.** Plastic strain energy dissipated by beams $W_{beam}$; summary of ratios between median values obtained from models MB, LB and HB under Ground Motion Set ESB binned by ADS and storey.

| | | | | Median $\hat{W}_{beam}$ | | | | | | Dispersion $W_{beam}^{84} - W_{beam}^{16}$ | | | | | |
| | | | Storey | MB/HB | | | LB/HB | | | MB/HB | | | LB/HB | | |
| Type | Amount | a | | D | C | ALL | D | C | ALL | D | C | ALL | D | C | ALL |
|---|---|---|---|---|---|---|---|---|---|---|---|---|---|---|---|
| ALL | ALL | 1 | ALL | 1.51 | 1.64 | 1.58 | 6.66 | 10.12 | 8.39 | 1.32 | 1.39 | 1.35 | 5.04 | 6.27 | 5.65 |
| ALL | ALL | 0.5 | ALL | 1.92 | 1.95 | 1.93 | 18.04 | 18.46 | 18.25 | 1.74 | 1.64 | 1.69 | 10.87 | 9.36 | 10.12 |
| ALL | 1 | ALL | ALL | 1.29 | 1.22 | 1.25 | 3.11 | 2.43 | 2.77 | 1.19 | 1.13 | 1.16 | 1.78 | 1.57 | 1.67 |
| ALL | 2 | ALL | ALL | 1.61 | 1.68 | 1.64 | 8.93 | 9.37 | 9.15 | 1.47 | 1.47 | 1.47 | 5.11 | 4.81 | 4.96 |
| ALL | 3 | ALL | ALL | 2.08 | 2.31 | 2.19 | 23.81 | 29.69 | 26.75 | 1.78 | 1.79 | 1.78 | 16.21 | 16.32 | 16.26 |
| Q | ALL | ALL | ALL | 1.51 | 1.52 | 1.51 | 7.07 | 10.04 | 8.55 | 1.28 | 1.25 | 1.27 | 5.02 | 5.94 | 5.48 |
| R | ALL | ALL | ALL | 1.55 | 1.67 | 1.61 | 9.34 | 11.97 | 10.65 | 1.46 | 1.50 | 1.48 | 6.24 | 6.85 | 6.54 |
| S | ALL | ALL | ALL | 2.08 | 2.19 | 2.14 | 20.65 | 20.87 | 20.76 | 1.84 | 1.78 | 1.81 | 12.61 | 10.66 | 11.63 |
| ALL | ALL | ALL | 1 | 1.48 | 1.15 | 1.32 | 4.92 | 1.83 | 3.38 | 1.24 | 1.07 | 1.16 | 1.96 | 1.14 | 1.55 |
| ALL | ALL | ALL | 5 | 1.56 | 2.06 | 1.81 | 17.68 | 36.80 | 27.24 | 1.51 | 1.98 | 1.75 | 13.54 | 22.69 | 18.12 |
| ALL | ALL | ALL | ALL | 1.71 | 1.79 | 1.75 | 12.35 | 14.29 | 13.32 | 1.53 | 1.51 | 1.52 | 7.96 | 7.82 | 7.89 |

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
