# Peer review of "Influence of Maxwell Stiffness in Damage Control and Analysis of Structures with Added Viscous Dampers"

_applsci, doi:10.3390/app11073089_

Round 1

Reviewer 2 Report

The paper deals with a numerical investigation on the effect of the Maxwell stiffness of fluid viscous dampers on the seismic performance of structures. The authors performed a comprehensive, well documented and well commented parametric study considering a case study planar steel moment resisting frame, and incorporating different assumptions of the brace stiffness.

Overall, the paper is an important contribution to the research field, is well structured, and excellently written. I recommend it for publication provided the following minor comments are addressed:

  • The sensitivity of the results to the brace stiffness may change depending on the power law exponent alpha. It seems that the authors only explored alpha=0.5 and alpha = 1.0, however manufacturing companies strive for achieving very low alpha values to maximize the energy dissipation capabilities (e.g., alpha =0.1 or even lower). Could the authors add some comments about the possible extensions of the conclusions drawn in this study to such lower range of alpha values?
  • An intricate aspect concerns the distribution of dampers along the building height and among the bays of the frame. The authors adopted three simplified distribution (uniform and triangular). There are more rational yet simplified procedures like the storey shear proportional distribution or the storey shear to efficient storeys distribution. The authors may wish to mention this important aspect by quoting relevant papers from the literature, such as the review paper:

De Domenico, D., Ricciardi, G., & Takewaki, I. (2019). Design strategies of viscous dampers for seismic protection of building structures: a review. Soil Dynamics and Earthquake Engineering, 118, 144-165.

Another related aspect, which is not investigated in this paper, is the distribution of the dampers among various bays, which may significantly alter the collapse mechanism because of concentration of axial forces in the columns. This aspect would deserve some limited comments from the authors.

  • The study is really extensive, and the reader may lose to catch the important conclusions of this paper. As a suggestion, it would be useful to emphasize more the key (quantitative) conclusions from this study, in the abstract and in the conclusions.

Congratulations on this interesting piece of work!
